# A slot region in the magnetosphere of Jupiter

**Minyi Long**[1,2], **Elias Roussos** [2] ✉, **Binbin Ni** [1,3] ✉, **Qianli Ma** [4,5] ✉,
**Peter Kollmann** [6], **Ruoxian Zhou**[7], **George Clark**[6], **Norbert Krupp** [2], **Xing Cao**[1],
**Peng Lu**[1], **Yixin Hao**[2] & **Shaobei Wang**[1]

Understanding the balance between charged particle acceleration and loss is central to radiation belt research. Jupiter's Galilean moons orbit within its intense radiation environment and can act both as sources and sinks of energetic particles. Using observations from the Juno spacecraft, we identify large-scale depletions of energetic electrons along Europa's orbit. These depletions are too deep to result from direct absorption by the moon alone. Here we show that rapid electron losses, occurring within a timescale shorter than Jupiter's rotation, are driven by pitch angle scattering via whistler-mode waves co-located with Europa's orbit. This suggests that Europa maintains a plasma environment capable of sustaining a slot-like region, similar to the one seen in Earth's Van Allen belts. However, this Jovian slot only partially extends along Europa's path, implying that additional, unidentified acceleration mechanisms may act to refill the region and maintain high radiation levels close to Jupiter.

The Van Allen radiation belts are regions of trapped high-energy charged particles within the terrestrial magnetosphere, which significantly affect satellites and other space-based technologies[1]. Understanding the formation, structure, and dynamics of these belts has been a key scientific focus for decades[2]. One notable feature within these belts is the slot region, a relatively low-radiation zone for electrons with energies above several hundred keV, located between the inner and outer electron radiation belts[3]. The slot region is formed as a result of a delicate balance between the inward transport of electrons and pitch angle diffusion driven by plasma waves[4].

Although it is well understood that the formation and maintenance of the slot region in Earth's radiation belts is primarily driven by the fundamental process of wave-particle interactions in geospace[2–4], the corresponding processes in Jupiter's magnetosphere remain much less explored. Jupiter's magnetosphere provides a unique plasma environment. It hosts extremely energetic particles[5,6] and complex plasma wave activities[7,8]. In particular, Jupiter's various natural moons act as both sources of plasmas and sinks for energetic particles[9,10], profoundly shaping the structure of the Jovian radiation belt and the broad magnetospheric environment[11–13]. The electrodynamic coupling between the moons and the rapidly rotating magnetosphere significantly alters the distributions of charged particles

and plasma waves[14–16]. Observations have suggested that enhanced electron anisotropies near the orbits of moons such as Europa and Ganymede should be responsible for intensifying whistler-mode waves[8,15]. These waves are believed to be key drivers of nonadiabatic electron acceleration and loss, playing an important role in regulating the energetic electron populations in planetary radiation belts[16,17].

However, previous studies have offered contradicting predictions about the exact role of wave-particle interactions in the particle variations of Jupiter's magnetosphere. For instance, some researchers[8] questioned whether enhanced whistler-mode waves at Ganymede and Europa would accelerate or deplete electrons. In contrast, some other studies[17] seem to favor electron losses over acceleration at energies below 1 MeV. Theoretical researches suggested that whistler-mode waves could effectively scatter electrons at energies from several keV to several hundreds of keV[17]. At Saturn's moons, such as Rhea, strong whistler-mode waves are excited by the electron loss-cone distributions, which may further contribute to spatially extended losses of energetic electrons observed in the vicinity of the moons[18,19]. A Juno-era report[20] documented the absence of 100's keV electrons near the magnetic equator within the Jovian M-shell range of 9–16 (the value of M-shell refers to the distance from the magnetic equator to Jupiter's center, normalized by Jupiter's radius $R_J$, where $R_J = 71{,}492$ km). In this

[1]School of Earth and Space Science and Technology, Wuhan University, Wuhan, China. [2]Max Planck Institute for Solar System Research, Göttingen, Germany. [3]CAS Center for Excellence in Comparative Planetology, Hefei, China. [4]Center for Space Physics, Boston University, Boston, MA, USA. [5]Department of Atmospheric and Oceanic Sciences, University of California, Los Angeles, CA, USA. [6]Johns Hopkins University Applied Physics Laboratory, Laurel, MD, USA. [7]Department of Physics, University of Texas at Dallas, Dallas, TX, USA. ✉e-mail: roussos@mps.mpg.de; bbni@whu.edu.cn; qianlima@atmos.ucla.edu

study, the M-shell parameter is calculated based on the JRM33 internal magnetic field model (order 13)[21] and the CON2020 current sheet model[22] to characterize the magnetic geometry in our analysis. While this absence of electrons possibly hints at the existence of a slot-like region of Jupiter's radiation belt, direct observational evidence of the slot formation in Jupiter's magnetosphere remains scarce.

In this work, we present observational and modeling evidence for a slot region in Jupiter's magnetosphere. By analyzing energetic electron fluxes and whistler-mode wave observations from Juno, we identify large-scale electron depletions near Europa's orbit that are not attributable to direct moon absorption. We show that these depletions are driven by pitch angle scattering from locally enhanced plasma waves, forming a slot-like structure in the magnetosphere of Jupiter. Our findings offer new insights into the dynamic processes that regulate Jovian radiation environments and support the existence of slot formation mechanisms similar to those observed at Earth.

## Results

### Observations of electron loss and whistler-mode waves near Europa's orbit

Previous studies using Galileo wave data have demonstrated the significant role of plasma waves in shaping Jupiter's magnetosphere, especially through wave–particle interactions that can lead to both particle acceleration and loss[23,24]. Juno spacecraft provides a unique opportunity to investigate the existence and formation mechanisms of slot-like regions in Jupiter's radiation belt. In this study, a gap of energetic electrons from tens of keV to hundreds of keV is observed to be accompanied by strong whistler-mode waves near Europa's orbit. The electrons at tens to hundreds of keV are the "seed" population for local acceleration to multiple MeV energies in the magnetospheres of Earth[25,26] and Jupiter[27]. Figure 1 illustrates a representative whistler-mode wave event and the corresponding distributions of energetic electron flux observed by Juno during one inbound trajectory on July 10, 2017. The whistler-mode waves were detected at the magnetic latitudes of 20–50° around Europa orbit (M-shell about 9.5) during the time interval marked by two vertical dotted lines. Following the previous work[28], we categorize the whistler-mode waves at $0.05f_{ceq} < f < f_{ceq}$ frequencies ($f$ is wave frequency, and $f_{ceq}$ is the equatorial electron gyrofrequency) as chorus or high frequency whistler-mode waves (HFW) and the waves at $f < 0.05f_{ceq}$ as hiss waves or low frequency whistler-mode waves (LFW) (Fig. 1a). LFWs with frequencies below $0.05f_{ceq}$ exhibited strong wave power, while HFWs above $0.05f_{ceq}$ displayed relatively weaker intensities. We calculate the whistler-mode wave amplitudes over the frequency range from 50 Hz to the equatorial electron gyrofrequency (see "Methods"). The whistler-mode wave amplitudes were mainly 30–70 pT with several instantaneous peaks reaching approximately 100 pT (Fig. 1f). During this period of the wave intensification, the fluxes of energetic electrons, ranging from tens of keV to several hundreds of keV, exhibited a noticeable decay (Fig. 1b–e). Notably, at 21:37 UT the Juno spacecraft passed the same magnetic field line (M-shell about 9.6) as that Europa crossed earlier at 20:00 UT (Supplementary Fig. 1). The decline in electron fluxes across the three selected energy channels spanned a wide range of local pitch angles. The increase of electron fluxes at 55 keV around UT 20:40 and UT 21:20 were likely caused by the electron injections[27] (Fig. 1b, c). This typical event indicates an observationally evident connection between the activity of whistler-mode waves and the energetic electron loss near Europa's orbit in Jupiter's magnetosphere.

### High correlation between energetic electron slot and whistler-mode wave peak near Europa

With the availability of Juno measurements spanning from perijove-03 (PJ-03) to perijove-46 (PJ-46), a statistical analysis is conducted to investigate the global distributions of energetic electron fluxes and whistler-mode waves in Jupiter's magnetosphere. Note that the Europa

flyby data during PJ-45 are excluded from our analysis to avoid any potential effect of moon proximity. The averaged whistler-mode wave amplitudes exhibit strong peaks around Europa's orbit along the magnetic field at both dawn and dusk local times, accompanied by a corresponding statistical decay in energetic electron fluxes (Supplementary Figs. 2 and 3). This indicates a clear spatial correlation between the intensification of whistler-mode waves and the loss of energetic electrons in Jupiter's radiation belt. Notably, the average wave amplitudes near Europa are stronger on the dawnside than on the duskside, corresponding to a clear depletion region for electron fluxes in the dawn sector. This local time asymmetry suggests that electron loss due to whistler-mode waves near Europa in the dusk sector may be replenished relatively quickly through radial or energy diffusion. Consequently, our analysis focuses on exploring the spatial correlation between energetic electron fluxes and whistler-mode wave amplitudes, specifically in the dawn sector (00–06 MLT) of Jupiter (Fig. 2).

In order to investigate the extent of flux decay in detail, the omni-directional electron fluxes ($j_{omni}$) are further normalized by the maximum $j_{omni}$ within each magnetic latitude bin for illustrative purpose. The normalized electron fluxes ($f_N$) for the three energy channels display a clear slot near Europa's orbit, spanning M-shell = 9–10 (Fig. 2a–c). Correspondingly, the average amplitudes of whistler-mode waves present an enhanced ribbon structure in the same region (Fig. 2d). Such a strong negative correlation is confirmed quantitatively by analyzing the latitudinally averaged profiles of whistler-mode waves and normalized electron fluxes as a function of M-shell (Fig. 2e). Specifically, the corresponding correlation coefficients between the two profiles over M-shell of 6.75–12.25 are −0.94, −0.87, and −0.65, and the corresponding $p$ values are $6.5 \times 10^{-6}$, $2.5 \times 10^{-4}$, and $2.2 \times 10^{-2}$ for each energy channel. The $p$ values are all less than 0.05, indicating that there is a statistically real and highly correlated relationship between whistler-mode wave amplitudes and electron flux variations at Europa's orbit. Hence, accumulated observations from Juno not only reveal an energetic electron slot in Europa's neighboring region but also imply a close linkage of the occurrence of this slot region to the presence of intense whistler-mode waves.

### Electron loss and slot formation driven by whistler-mode waves near Europa

To quantify the specific contribution of whistler-mode waves to the formation of observed slot of energetic electrons from tens of keV to hundreds of keV, the wave distribution model is developed based on the statistically averaged wave power spectral densities obtained from Juno measurements (Fig. 3a–c). Overall, the average power of LFWs is about one order of magnitude stronger than that of HFWs over a broad M-shell range. LFWs show small variations with the magnetic latitude, maintaining pronounced wave power up to $|MLAT| = 60°$. Interestingly, there exists a distinct wave power peak of LFWs around Europa. In contrast, HFWs within $|MLAT| < 20°$ exhibit a larger average wave power by an order of magnitude, compared to those at $|MLAT| > 20°$.

In this study, the averaged wave power densities observed within $|MLAT| < 20°$ at M-shell = 9.5 (Fig. 3c) are used as inputs of LFWs and HFWs for quantifying the wave-driven electron scattering effects. It is further considered that LFWs ($<0.05f_{ceq}$) are present over the magnetic latitudes of 0–60° and that HFWs ($>0.05f_{ceq}$) are confined within the latitudes below 20°. LFWs and HFWs have different scattering effects on Jovian energetic electrons, according to the calculated bounce-averaged diffusion coefficients (Fig. 3d, e and Supplementary Fig. 5; see Methods for details of diffusion rate computation). LFWs primarily scatter electrons with energies >100 keV via cyclotron resonance at equatorial pitch angles ($\alpha_{eq}$) <75° but can also resonate with electrons from about 10 to 1 MeV at higher pitch angles ($\alpha_{eq} > 80°$) through Landau resonance (Fig. 3d). In contrast, HFWs predominantly scatter electrons at energies of 10 –100's keV through cyclotron resonance (Fig. 3e). In combination, LFWs and HFWs can drive efficient

diffusion of Jovian radiation belt electrons with energies of 10–1 MeV over a wide range of pitch angle (Fig. 3f). The net bounce-averaged pitch angle diffusion rates are 1–2 orders of magnitude greater than the mixed and momentum diffusion coefficients, and can approach the level of several $10^{-5}\,s^{-1}$ (Supplementary Figs. 5 and Fig. 3f). This brings estimates of electron loss timescale of several hours, indicating that the dominant contribution of whistler-mode waves should be pitch angle scattering accounting for electron loss in Jupiter's radiation belt.

With the net bounce-averaged diffusion coefficients at M-shell = 9.5 (Fig. 3f and Supplementary Fig. 5h, i) as inputs, the temporal evolution of phase space densities (PSDs) of Jovian energetic electrons is simulated by numerically solving the two-dimensional Fokker-Planck diffusion equation (see "Methods" for details of the model). The

statistical electron PSD distribution at this location (Fig. 3g) is adopted as the initial condition for the simulations. The simulated results (Fig. 3h, i) clearly indicate that under the impact of whistler-mode wave-induced scattering, the energetic electron PSDs decay gradually and their pitch angle distributions evolve as a whole to reproduce very well the measured PSD distributions within 10 h. The simulation results show reasonable consistency with the observations, with the ratios of the averaged electron PSD between the simulations and observations for the three indicated energy channels being approximately 0.95, 0.95, and 0.98, indicating that the model captures the overall trend of the measured electron fluxes. This good agreement between the modeled and observed results justifies the significant role of whistler-mode waves in driving electron loss in Jupiter's radiation belt and in producing the slot region near Europa. It is noted that the simulated

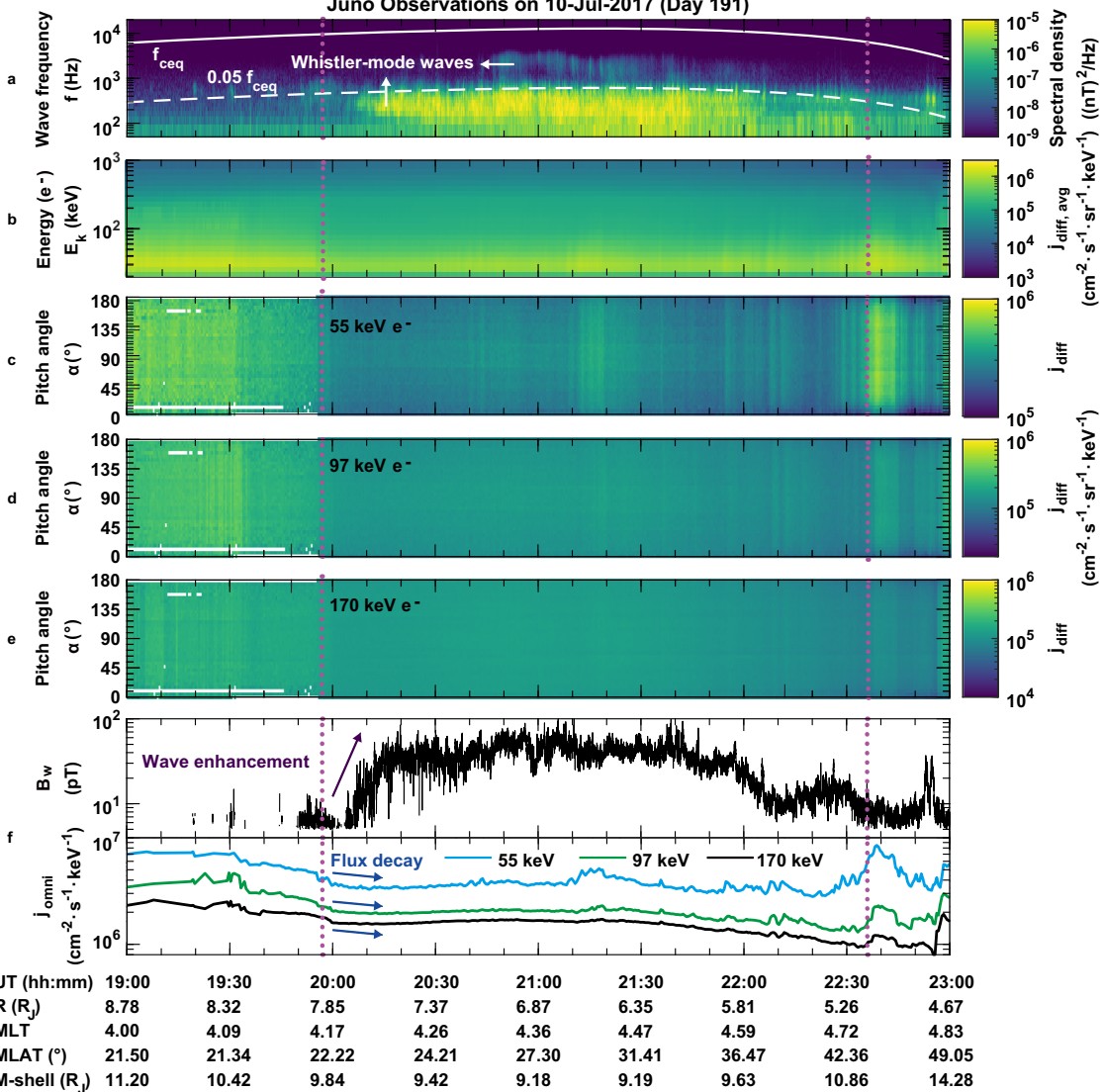

**Fig. 1 | Simultaneous observations of the energetic electron flux decay and whistler-mode wave enhancement near Europa's orbit on July 10, 2017. a** The frequency-time spectrogram illustrating the magnetic field power spectral density as observed by the WAVES instrument[43]. The solid line marks the equatorial electron gyrofrequency ($f_{ceq}$), while the dashed line indicates $0.05f_{ceq}$. **b** energy-time spectrograms of averaged electron differential fluxes ($j_{diff,\ avg}$) recorded by the Jupiter Energetic Particle Detector Instruments (JEDI)[29]. **c–e** the pitch angle distributions of differential electron flux ($j_{diff}$) for three selected electron energies (55, 97, and 170 keV). **f** the 30 s averaged amplitude of whistler-mode waves ($B_w$) (upper

panel) and the omni-directional electron fluxes ($j_{omni}$) in the lower panel across the three different energy channels: 55 keV, 97 keV, and 170 keV. Vertical dashed lines mark the interval of interest. The equatorial magnetic field intensity, M-shell (the distance from the magnetic equator to Jupiter's center, normalized by Jupiter's radius $R_J$, where $R_J = 71,492$ km), magnetic latitude (MLAT), and magnetic local time (MLT) are calculated using the JRM33 internal magnetic field model (order 13)[21] and the CON2020 current sheet model[22]. UT refers to universal time, and $R$ denotes the distance normalized to $R_J$ from Juno to the center of mass of Jupiter. Source data are provided as a Source Data file.

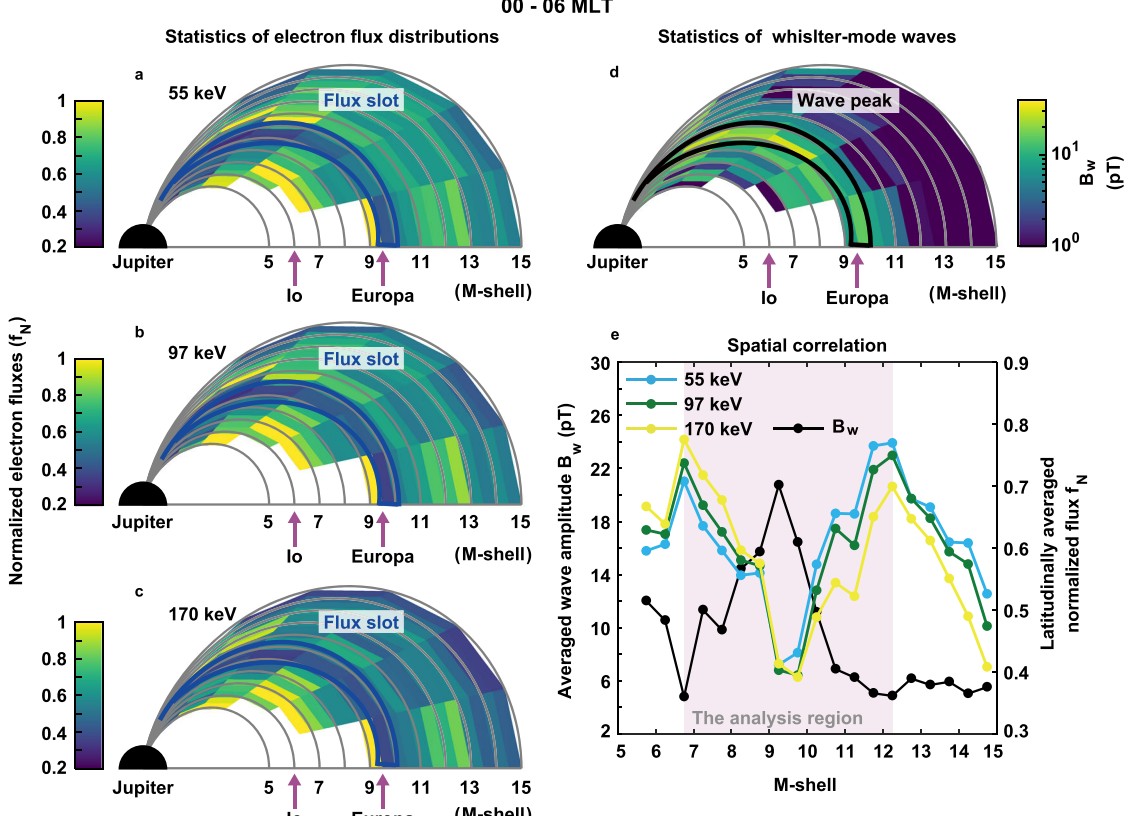

**Fig. 2 | Spatial correlation of energetic electron fluxes and whistler-mode wave amplitudes ($B_w$) in the neighboring environment of Europa.** MLAT denotes magnetic latitude, MLT means magnetic local time, and the M-shell is defined as the distance from the magnetic equator to Jupiter's center, normalized by Jupiter's radius. **a–c** the normalized electron fluxes in the magnetic meridian plane (Δ|MLAT| = 10°) for the three specified energy channels (55, 97, 170 keV) within the equivalent M-shell range of 5–15 (ΔM-shell = 0.5) in the dawn sector (MLT = 00–06). The normalized flux ($f_N$) is the ratio between the omni-directional electron fluxes ($j_{omni}$) and the maximum $j_{omni}$ within each magnetic latitude bin within 5 < M-shell < 15. **d** the averaged whistler-mode wave amplitudes in the magnetic meridian plane on the dawnside. Flux slot indicates the region of reduced electron fluxes, while wave peak highlights the region with enhanced wave amplitudes. **e** Spatial correlation between latitudinally averaged normalized electron fluxes for the three energy channels and averaged wave amplitudes, calculated within the shaded region of M-shell = 6.75–12.25 (shaded region). Due to limited data coverage within M-shell <9 within |MLAT| < 10°, the latitudinally averaged normalized electron fluxes and wave amplitudes are calculated within |MLAT| = 10–60°. Source data are provided as a Source Data file.

electron PSDs tend to underestimate the observed levels at lower pitch angles (i.e., <30° or >150°), possibly due to undetermined field-aligned electron sources or limited electron measurements near the equatorial loss cone[29]. Despite this, whistler-mode waves at Jupiter are efficient to remove the population of energetic electrons on timescales of hours, therefore critically shaping the Jovian radiation belt slot near Europa's orbit.

Furthermore, the simulated energy spectrum of Jovian energetic electrons aligns closely with the observed shape after 10 h of wave-particle resonant interactions (Fig. 4a–c). Although the first thought might be that this energetic electron slot could be caused by absorption of Europa[10,11], our investigation demonstrates that it is mainly attributed to wave-induced scattering in the vicinity of the moon, in a manner similar to the depletion of MeV protons driven by ion cyclotron waves near Io[12]. A single moon passage, for example Europa, primarily creates time- and longitude-dependent particle depletion in the wake of the moon (i.e., microsignatures)[30], which will be then refilled by fast inward radial diffusion. In addition, these microsignatures have the scale of a moon's diameter, which corresponds to 0.022 $R_J$ for Europa, much smaller than the depletion scales of the electron slot observed with Juno (Figs. 1c–e and 2a–c). The broader, longitude- and time-averaged effects of moon absorption on the radial particle distribution occur after multiple orbits, when losses from different microsignatures add up (i.e., macrosignatures)[30]. However, these

macrosignatures remain relatively weak near Europa's orbit due to a relatively efficient radial diffusion process expected at Europa's distance[12].

To better distinguish the specific contributions of wave-induced scattering and Europa's absorption to electron loss, the radial profiles of 97 keV electron fluxes within M-shell = 7–12 are simulated using the radial diffusion equation with loss terms (Fig. 4d; see "Methods" for details of the model). Under the influence of radial diffusion and Europa's absorption, the omni-directional fluxes of Jovian energetic electrons decrease monotonically with increasing M-shell, showing no signature of a slot but a weak decay near M-shell = 9.5 (dashed curve). In contrast, inclusion of whistler wave-driven scattering generates very pronounced electron loss in the vicinity of Europa to form a slot region (solid curve). Moreover, by further increasing the radial diffusion rates by 5 times, the energetic electron slot can be smoothed quickly by faster inward transportation (dotted curve). As a consequence, it is evident that the formation of the Jovian radiation belt slot and its evolution is a delicate interplay of different physical processes at Jupiter, including wave-particle interactions[8,17], moon absorption[10,11], radial diffusion[7], and even possible electron injections[27]. Our simulations highlight the dominant role of wave-particle interactions in forming the Jovian radiation belt slot, which can be readily incorporated into existing particle dynamic models that so far focus solely on moon absorption in Jupiter's magnetosphere.

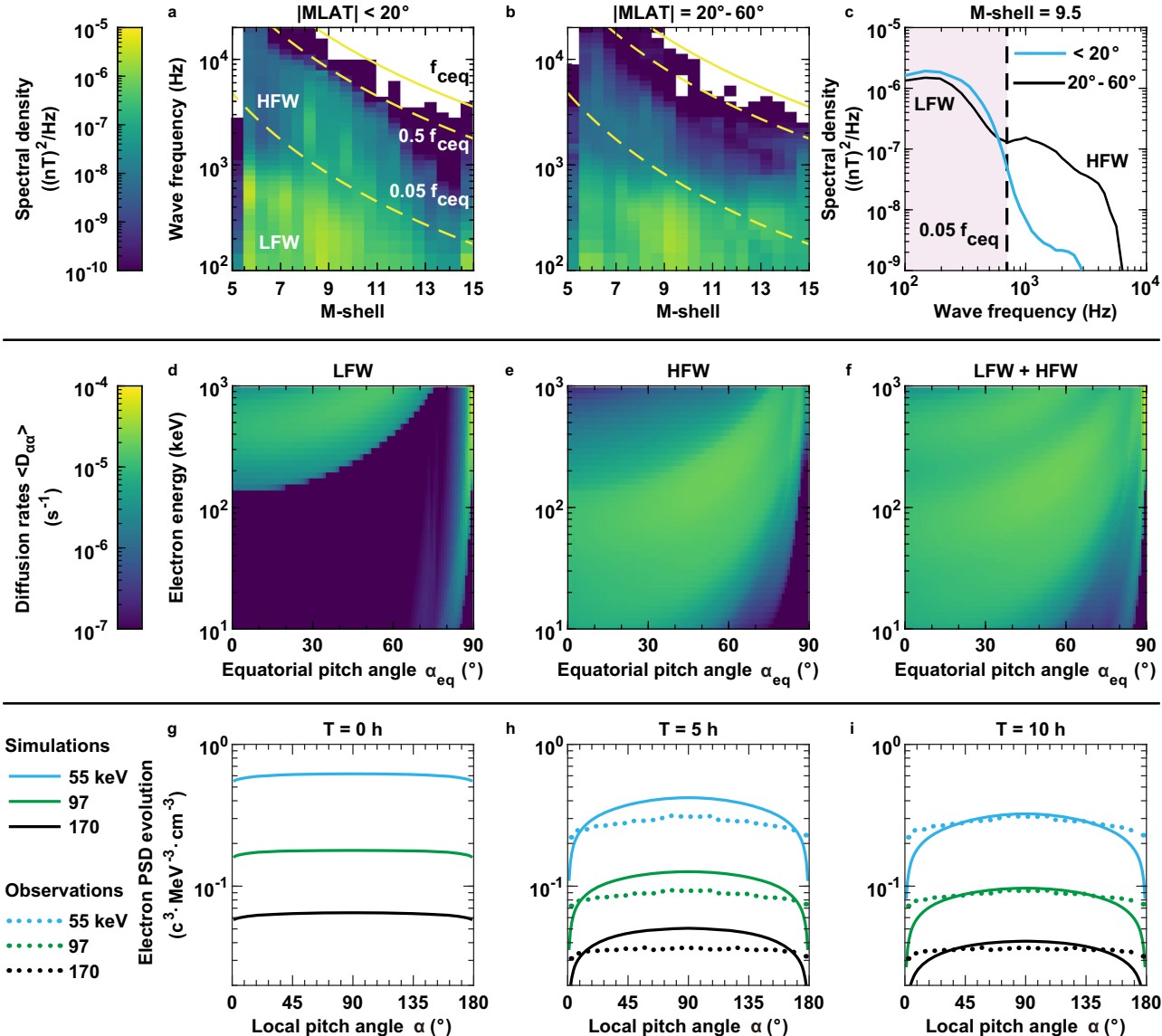

**Fig. 3 | Spectral distributions of Whistler-mode waves and their quantitative scattering impact on Jovian radiation belt energetic electrons. a, b** the time-averaged wave intensities as functions of wave frequency and M-shell (the distance from the magnetic equator to Jupiter's center, normalized by Jupiter's radius) for |MLAT| < 20° (MLAT is the magnetic latitude) and |MLAT| = 20–60°. From top to the bottom, the solid line and two dashed lines represent $f_{ceq}$ (equatorial electron gyrofrequency), $0.5f_{ceq}$, and $0.05f_{ceq}$ based on the dipole magnetic field model. **c** the average wave intensity at M-shell = 9.5 for |MLAT| < 20° and |MLAT| = 20–60°. The shaded region indicates the frequency range of LFWs (low frequency whistler-mode waves), while the unshaded part corresponds to HFWs (high frequency whistler-mode waves). **d, e** the bounce-averaged pitch angle diffusion coefficient $<D_{\alpha\alpha}>$ induced by LFWs and HFWs as functions of energy and equatorial pitch angles. **f** the combined $<D_{\alpha\alpha}>$ from LFWs and HFWs. **g–i** Comparison of simulated electron pitch angle distributions (solid lines) with observations (dotted lines) for three energy channels (55, 97, 170 keV) at three simulation intervals (0, 5, and 10 h). The observed PSDs (phase space densities) are from the time interval between UT = 8:00 and UT = 8:30 (UT means universal time) in Fig. 1, excluding the interval of electron injections. Source data are provided as a Source Data file.

## Discussion

The observations and simulations in the present study provide the evidence demonstrating that a slot-like region exists in Jupiter's electron radiation belt and that strong whistler-mode waves near Europa's orbit play a dominant role in forming this slot. A scenario is schematically illustrated in Fig. 5 to explain such a closely connected and coupled system between Jupiter's magnetosphere and Europa. Firstly, the temperature anisotropy of hot electrons injected via interchange instability provides the free energy necessary for the excitation of whistler-mode waves over a broad region in Jupiter's magnetosphere. HFWs and LFWs can resonate with approximately 1–50 keV electrons and >about 50 keV electrons, respectively, over the nearly entire pitch angle range. The source electron population at equatorial pitch angles of about 60–90° with anisotropic distributions is significant to the

wave generation and amplification. It has been shown that injected electrons with typical energies of a few keV to approximately 50 keV and occasionally reaching above 300 keV, exhibiting temperature anisotropies, are capable of exciting whistler-mode chorus waves under typical Jovian magnetospheric conditions[27]. The hot electron angular distribution can be further affected by depletions at low equatorial pitch angles due to Europa's absorption[10,11] and by losses within neutral materials (e.g., the gas torus) due to scattering or energy dissipation through friction[31–33]. Strong temperature anisotropies induced by these materials have been observed in protons near Europa[32] and may similarly occur in electrons. The enhanced hot electron temperature anisotropies induced by these processes supply extra free energy for the growth of whistler-mode waves, amplifying their intensities around Europa's orbit[8]. Consequently, intense

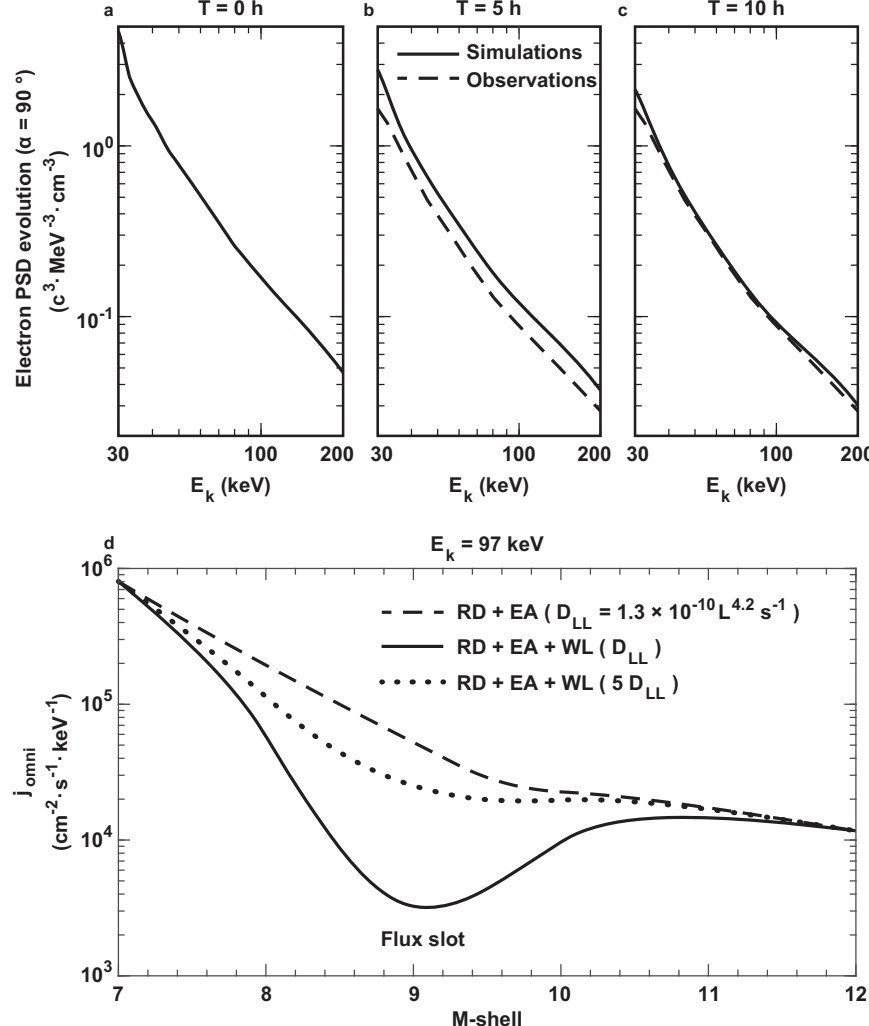

**Fig. 4 | Formation of an energetic electron slot in the vicinity of Europa under considerations of the contributions from different physical processes.**
**a**–**c** Comparisons between simulations (solid lines) and observations (dashed lines) of the electron energy spectra from 30 to 200 keV at a pitch angle of 90° for three different simulation intervals (0, 5, and 10 h). M-shell denotes the distance from the magnetic equator to Jupiter's center, normalized by Jupiter's radius, and $E_k$ is electron energy. **d** the dashed line represents the omnidirectional electron fluxes ($j_{omni}$) of electrons when considering the steady-state solution of the radial diffusion (RD) equation for 97 keV electrons, incorporating the absorption effect of Europa (EA). The solid line shows the result after including Europa absorption and wave loss (WL). The dotted line is the steady-state solution when the radial diffusion rates increase by 5 times, and the value of radial diffusion rate ($D_{LL}$) refers to the electron radiation model[12]. The Europa absorption region spans M-shells 9.29–10.02, which is from the geometrical calculation based on the magnetic field model for Jupiter[21,22]. The spatial coverage of wave-induced loss falls within M-shells of 8–10, following the statistical results in Fig. 2. Source data are provided as a Source Data file.

whistler-mode wave emissions can strongly scatter Jovian energetic electrons for atmospheric loss through the pitch angle diffusion process, leading to the formation of an electron slot—an analog to the two-zone Van Allen radiation belts around Earth[1]. The resultant precipitated electrons can also produce diffuse auroral emissions in Jupiter's upper atmosphere[34], in a manner similar to that at Earth[35,36]. These results reveal a first comprehensive demonstration of the significant contribution of whistler-mode waves near Europa to the formation of a Jovian radiation belt slot in a global context. Our findings can lay a useful baseline for future studies of radiation belts around other magnetized planets with satellites.

Noteworthily, while the present analyses focus mainly on the slot and wave characteristics in the dawn sector of Jovian radiation belt, a statistical dawn-dusk asymmetry has been captured observationally in the average amplitudes of whistler-mode waves, with the corresponding electron slot near Europa being more pronounced in the dawn sector (Supplementary Figs. 2 and 3). Multiple dawn-dusk asymmetries have been reported within Jupiter's magnetosphere,

especially the corresponding charged particle distributions[37–41]. In addition, the formation of the electron slot is characterized by a limited extent in magnetic local time, with the most prominent depletion observed in the dawn sector. This feature indicates that the slot is predominantly strongly influenced by the MLT-dependent wave activity. While our current simulation can explain the general formation of the energetic electron slot under the impact of whistler-mode waves near Europa, further analysis and modeling attempts are needed to fully understand the observations of this slot region at Jupiter. In the absence of strong whistler-mode wave fields, the remaining electron loss processes due to Europa absorption and energy losses in the moon's neutral torus are insignificant on a global scale. Furthermore, the slot region can be refilled within the few hours that electrons need to drift between dawn and dusk. This fact points to either very fast electron radial diffusion or the presence of a yet unidentified acceleration processes that may quickly refill the depleted electrons and sustain high radiation levels near or inward of Europa's orbit, which eventually power the Jupiter's radiation belts.

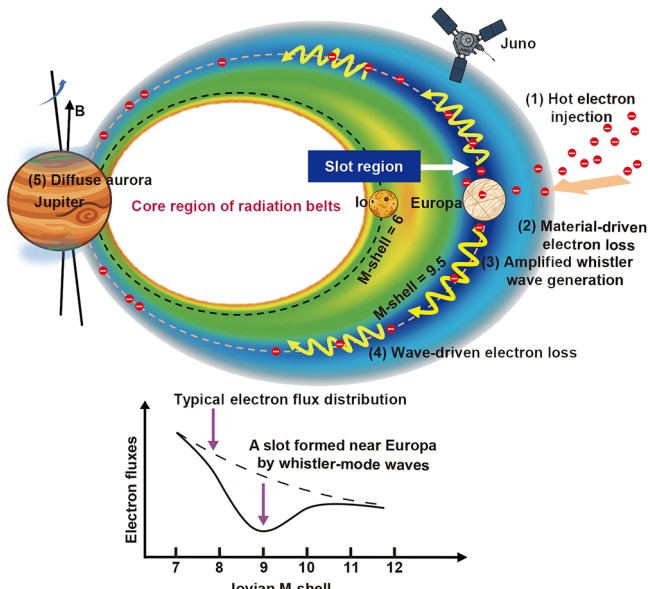

**Fig. 5 | A schematic diagram illustrating the formation of the slot region near Europa within the closely connected system of Jupiter and its moons.** Injected hot electrons due to interchange instability exhibit anisotropic pitch angle distributions, which provide the free energy necessary to excite whistler-mode waves within Jupiter's magnetosphere. Material-driven losses at Europa's orbit (such as moon absorption process, or loss by neural torus) will deplete electrons with low equatorial pitch angles to enhance electron anisotropies, leading to the generation of more intense whistler-mode waves. These increased waves efficiently scatter energetic electrons into Jupiter's atmosphere, playing a dominant role in forming the electron slot region near Europa's orbit and producing diffuse aurora. Source data are provided as a Source Data file.

## Methods

### Distributions of electron fluxes and whistler-mode waves in Jupiter's radiation belt

The Juno data from perijove-03 (PJ-03) to perijove-46 (PJ-46) are used for analyzing the global distributions of energetic electrons and whistler-mode waves in Jupiter's magnetosphere. The energy and pitch angle distributions of electrons over energy ranges from about 30 keV to approximately 1 MeV are provided by Jupiter Energetic Particle Detector Instruments (JEDI)[29]. Observed electron distributions at different energies are obtained in every 30 s within every 5° pitch angle bin. The electron energy spectra were corrected according to the method proposed by Mauk et al.[42] by removing artifacts caused by minimum ionizing peaks (about 100–200 keV) that result from the contamination by penetrating particles. In addition, we merge the flux data with low and high energy resolution and then pick up three electron energy channels (55, 97, and 170 keV) for our analysis. The averaged differential fluxes ($j_{\mathrm{diff,\,avg}}$) are calculated as in Eq. (1).

$$j_{\mathrm{diff,\,avg}} = \frac{\sum_{i=1}^{n} j_{\mathrm{diff}}^{i}}{n}, \tag{1}$$

where $j_{\mathrm{diff}}$ is the electron differential flux, and $n$ is the number of observation points. The omni-directional fluxes ($j_{\mathrm{omni}}$) are as shown in Eq. (2).

$$j_{\mathrm{omni}} = 2\pi \int_{0}^{\pi} j_{\mathrm{diff}} \sin \alpha \, d\alpha, \tag{2}$$

where α is the local pitch angle.

To investigate the statistics of whistler-mode waves in Jupiter's magnetosphere, we utilize electric and magnetic field power spectral

density data within the 50–20 kHz frequency range under the survey mode, as recorded by the Juno/Waves instrument[43]. The local magnetic field measurements provided by the Juno magnetometer[44] are mapped to the magnetic equator to estimate the equatorial field strength ($B_{\mathrm{eq}}$) along the same field line using the JRM33 + CON2020 model[21,22]. The following robust criteria are applied to select events of whistler-mode waves:

(1). Since we are interested in the spatial correlation between whistler-mode waves and Jupiter's moons, we consider whistler-mode waves in the range of M-shell = 5–15 and |MLAT| < 60°.

(2). The wave spectra observed with the frequencies between 50 Hz and the equatorial electron gyrofrequency ($f_{\mathrm{ceq}}$) are analyzed to cover a broad frequency range for whistler-mode wave identification.

(3). To capture realistic wave emissions, the magnetic field power spectral densities for each frequency channel should exceed the maximum between the threshold value of $10^{-8}$ nT²/Hz and the daily median value of observed power spectral densities.

(4). To exclude isolated emissions that are difficult to distinguish the properties, the selected events of whistler-mode waves should last more than 10 min per day, with each wave sample covering useful power spectral density data for at least three frequency channels.

(5). The wave amplitudes are calculated using the trapezoidal integration method. The data points with the computed wave amplitudes <5 pT are excluded since it is difficult to distinguish these weak wave emissions from the background noise level.

(6). All the wave amplitude data, originally with a 1 s resolution, are binned into a 30-s resolution to reduce the size of the entire dataset, which however, has little effect on the analysis results.

### Wave-induced diffusion rate calculations in Jupiter's JRM33 + CON2020 magnetic field configuration

In order to calculate the quasi-linear bounce-averaged electron diffusion coefficients driven by whistler-mode waves in Jupiter's radiation belt, we adopt the background plasma density model following the work of Shprits et al.[45]. Specifically, the plasma density $N(L, \lambda)$ ($\lambda$ is the magnetic latitude) is given by

$$N(L, \lambda) = N_{\mathrm{eq}}(L) \, e^{\frac{-L^2(1-\cos^6\lambda)}{3H^2}}, \tag{3}$$

Where the equatorial plasma density ($N_{\mathrm{eq}}$, unit of cm⁻³) is calculated by

$$N_{\mathrm{eq}} = 3.2 \times 10^8 L^{-6.9}, \tag{4}$$

and $H$ is the scale height. The value of $H$ is calculated by

$$H = 10^{a_1 + a_2 r + a_3 r^2 + a_4 r^3 + a_5 r^4}, \tag{5}$$

where,

$$r = \log_{10}(L), \tag{6}$$

The parameters are set as:

$$a_1 = -0.116, a_2 = 2.14, a_3 = -2.05, a_4 = 0.491, a_5 = 0.126. \tag{7}$$

The $L$ value is approximately equal to the value of Jovian M-shell.

Ray tracing results show that high-frequency whistler-mode waves generated near the magnetic equator of Jupiter's magnetosphere are typically confined within |MLAT| < about 20°, while low-frequency waves can reach higher latitudes[46]. As these waves propagate, their wave normal angles become increasingly oblique due to the effects of field line geometry, plasma density gradient and wave refraction[28,46]. Based on the statistics of whistler-mode wave distribution in Jupiter's

radiation belt, we assume that HFWs are confined within |MLAT| < 20° and LFWs are present up to |MLAT| = 60°. Different models of wave normal angle distribution are adopted to evaluate the electron diffusion rates induced by LFWs and HFWs separately. We follow a number of previous studies[47–49] to set the wave normal angle ($\psi$) distribution of whistler-mode waves as follows,

$$g(X) = \exp\left[\frac{-(X - X_m)^2}{(\delta X)^2}\right], \ X_{lc} \leq X \leq X_{uc}, \tag{8}$$

where,

$$X = \tan(\psi), \ \delta X = \tan(\delta\psi), \ X_m = \tan(\psi_m) X = \tan(\psi), \tag{9}$$

$$X_{lc} = \tan(\psi_{lc}), \ X_{uc} = \tan(\psi_{uc}). \tag{10}$$

Here, $\delta X$ is the angular width of the distribution, $X_m$ is the peak value of the wave normal angle distribution. $X_{lc}$ and $X_{uc}$ represent the lower and upper bounds of the wave normal angle distribution, respectively. The bounds ($X_{lc} \leq X \leq X_{uc}$) indicate that the distribution is limited to the interval between $X_{lc}$ and $X_{uc}$, outside of which the wave power is zero. The parameters adopted for the wave normal angle model are shown in Supplementary Table 1.

Following the diffusion rate computation equations in a non-dipolar magnetic field topology[17,50], we combine the JRM33 + CON2020 magnetic field configuration[21,22] and the statistically averaged whistler-mode wave power spectral intensity to quantify wave-induced bounce-averaged electron diffusion coefficients as a function of kinetic energy and equatorial pitch angle. The wave-particle interaction effects of both cyclotron resonances with the resonance order from −10 to 10 and Landau resonance are considered in the electron diffusion rate calculations.

## Fokker-Planck diffusion simulations in Jupiter's JRM33 + CON2020 magnetic field configuration

In order to simulate the evolution of electron phase space density under the impact of whistler-mode waves at the Jovian M-shell = 9.5 in the JRM33 + CON2020 magnetic field configuration[21,22], we solve numerically the two-dimensional (2-D) bounce-averaged Fokker–Planck diffusion equation as below,

$$\begin{aligned}\frac{\partial f}{\partial t} = &\frac{1}{Gp}\frac{\partial}{\partial \alpha_{eq}}\left[G\left(\langle D_{\alpha\alpha}\rangle\frac{1}{p}\frac{\partial f}{\partial \alpha_{eq}} + \langle D_{\alpha p}\rangle\frac{\partial f}{\partial p}\right)\right] \\ &+ \frac{1}{G}\frac{\partial}{\partial p}\left[G\left(\langle D_{p\alpha}\rangle\frac{1}{p}\frac{\partial f}{\partial \alpha_{eq}} + \langle D_{pp}\rangle\frac{\partial f}{\partial p}\right)\right],\end{aligned} \tag{11}$$

where $f$ is the electron phase space density, $p$ is the electron momentum, $\alpha_{eq}$ is the equatorial pitch angle. The value of $G$ is given by

$$G = p^2 T(\alpha_{eq}) \sin\alpha_{eq} \cos\alpha_{eq}, \tag{12}$$

with $T(\alpha_{eq})$ as the normalized bounce period in a non-dipole magnetic field[17,50]. $\langle D_{\alpha\alpha}\rangle$,$\langle D_{\alpha p}\rangle$, and $\langle D_{pp}\rangle$ denote the bounce-averaged pitch angle diffusion rate, momentum diffusion rate and cross diffusion rate, respectively. The two cross diffusion terms, i.e., $\langle D_{\alpha p}\rangle$ and $\langle D_{p\alpha}\rangle$, are equal to each other. These cross terms are included in our numerical diffusion simulations.

The initial condition of the electron PSD distribution function is obtained based on Juno measurements. Firstly, we convert observations of electron differential fluxes to electron PSD in terms of the following formula[51],

$$f = 3.33 \times 10^{-8}\frac{j_{diff}}{E_{MeV}(E_{MeV} + 2m_0 c^2)}, \tag{13}$$

where $E_{MeV}$ is the electron kinetic energy in unit of MeV, $m_0 c^2$ (approximately 0.511 MeV) is the electron rest energy with $m_0$ as the electron mass and $c$ the speed of light. The electron differential flux ($j_{diff}$) has a unit of cm$^{-2}$ s$^{-1}$ sr$^{-1}$ keV$^{-1}$, and the electron phase space density has a unit of c$^3$ MeV$^{-3}$ cm$^{-3}$. We acquire the initial electron PSD distributions near the magnetic equator based on the electron flux measurements from Jovian Auroral Distributions Experiment (JADE)[52] (about 100 eV to approximately 100 keV) and JEDI-E[29] (about 30 keV to approximately 1 MeV). We analyze the statistical distributions of electron fluxes near the magnetic equator (|MLAT| ≤ 5°) at M-shell = 9.5, similar to the method described in the work of Ma et al.[51], which are used as the initial conditions for the simulations. Specifically, the data from July 2016 to December 2023 are used, and the data are binned in 0.5 M-shell intervals. Regarding the set-up of boundary conditions, we assume that the electron PSD remains constant at both the lower and upper energy boundaries (i.e., $E_k = 10$ keV and 1 MeV). Furthermore, the boundary conditions for $\alpha_{eq}$ are defined as:

$$f(\alpha_{eq} \leq \alpha_{LC}) = 0, \tag{14}$$

$$\frac{\partial f}{\partial \alpha_{eq}}(\alpha_{eq} = 90°) = 0, \tag{15}$$

A hybrid finite difference method[53] is implemented to solve the 2-D bounce-averaged Fokker–Planck equation. We use fully implicit numerical method to solve the pitch angle diffusion and momentum terms, and an alternating implicit numerical method to solve the mixed terms. Such a combination of solving methods is called "hybrid". We simulate the temporal evolution of Jovian electron PSD under the impact of HFW and LFW within 10 h. To facilitate quantitative comparisons of the numerical results with Juno observations, the simulated electron PSDs as a function of $\alpha_{eq}$ (equatorial pitch angle) are transformed into a function of $\alpha$ (local pitch angle), following the conversion equation

$$\alpha_{eq} = \arcsin\left(\sqrt{\frac{B_{eq}}{B_o}\sin^2\alpha}\right), \tag{16}$$

where $B_o$ is the local magnetic field strength observed by Juno and $B_{eq}$ is the equatorial magnetic field strength. $B_{eq}$ is computed by mapping $B_o$ along the field line to the magnetic equatorial plane under the JRM33 internal magnetic field model (order 13)[21] and the CON2020 current sheet model[22].

## Radial diffusion simulations of steady-state electron distribution with incorporation of loss processes

To elucidate the physical mechanisms responsible for the formation of an electron flux slot near Europa's orbit, we analyze the steady-state solution of the radial diffusion equation[54], with incorporation of electron loss effects due to moon absorption and wave-particle interactions. The radial diffusion equation including the loss terms is described below,

$$\frac{\partial f}{\partial t} = L^2\frac{\partial}{\partial L}\left(L^{-2}D_{LL}\frac{\partial f}{\partial L}\right) - \frac{f}{\tau_w} - \frac{f}{\tau_m}. \tag{17}$$

Here, $L$ is the Jovian magnetic shell, which is approximately equivalent to the M-shell in a dipole magnetic field. $D_{LL}$ is the radial diffusion coefficient, which has been widely used for studying both

Earth's and Jupiter's electron radiation belts. It is computed by the following formula[12]

$$D_{LL} = 1.3 \times 10^{-10} L^{4.2}. \tag{18}$$

The term $\tau_m$ denotes the average electron lifetime due to Jupiter's moon absorption. Following the work of Long et al.[10], it is determined by

$$\tau_m = \frac{\tau_{enc}}{-\ln(1 - P_a)}, \tag{19}$$

where $\tau_{enc}$ is the average encounter time between a moon and energetic electrons near the moon's orbit, and $P_a$ is the averaged absorption probability per encounter, influenced by factors such as electron energy, pitch angle, the moon's size and orbit, and the background magnetic field[10,11]. The term $\tau_w$ is the electron loss time scale due to pitch angle scattering by Jovian whistler-mode waves, which can be calculated by ref. 55

$$\tau_w = \int_{\alpha_{LC}}^{\pi/2} \frac{1}{2\langle D_{\alpha\alpha} \rangle \tan \alpha_{eq}} d\alpha_{eq}. \tag{20}$$

Based on the simulation results in the JRM33 + CON2020 magnetic field configuration[21,22], the L-shell region of Jupiter's moon absorption is between 9.29 and 10.02. In contrast, the Jupiter's electron slot region induced by whistler-mode waves is mainly located within M-shell = 8–10. For simplicity, we assume that $\tau_w$ is constant for this spatial M-shell coverage and is derived using the electron pitch angle diffusion rates at M-shell = 9.5. By considering that the radial diffusion timescale of tens of days (Eq. (18)) is significantly longer than that of electron pitch angle scattering loss due to whistler-mode waves (about several hours) but comparable to the collisional loss timescale induced by Europa (about tens of days), we further set that the electron PSD remains constant at both the lower (M-shell = 7) and upper M-shell boundaries (M-shell = 12) so that the effect of wave-driven loss can be reasonably assessed. By adopting the initial electron fluxes given by the D&G electron flux model[56], we finally calculate the steady-state solutions of electron PSDs as a function of radial M-shell by setting the time derivative of the distribution function to zero and including/excluding different physical processes in Eq. (17), the results of which are shown in Fig. 4d for 97 keV electrons for illustrative purposes. All processed data used in this study are available at Figshare[57], ensuring full reproducibility of the results presented.

## Data availability
All Juno data used in this study were obtained from the Level 3 calibrated data available in the NASA Planetary Data System (PDS). The wave data used in this study are publicly available from the https://pds-ppi.igpp.ucla.edu/data/JNO-E_J_SS-WAV-3-CDR-SRVFULL-V2.0/. The background magnetic field data are available from the link: https://pds-ppi.igpp.ucla.edu/data/JNO-J-3-FGM-CAL-V1.0/. The JADE-E electron data are available from https://pds-ppi.igpp.ucla.edu/data/JNO-J_SW-JAD-3-CALIBRATED-V1.0/. The JEDI-E electron data are available from the link: https://pds-ppi.igpp.ucla.edu/data/JNO-J-JED-3-CDR-V1.0/. The source data generated in this study and underlying all figures in the main text and Supplementary Information have been deposited in Figshare (accession code: https://doi.org/10.6084/m9.figshare.29618030.v3). Additional datasets generated and/or analyzed during the current study are available from the corresponding author upon request. Source data are provided with this paper.

## Code availability
Custom MATLAB scripts used for spacecraft data processing, figure generation, and the G&D electron flux model have been deposited in Figshare: https://doi.org/10.6084/m9.figshare.29618030.v3. The JRM33

internal magnetic field model and the CON2020 current sheet model are available via the link: https://github.com/mattkjames7/JupiterMag. The moon absorption model is available at Figshare: https://doi.org/10.6084/m9.figshare.16557189.v6. The equations used to compute wave-induced diffusion coefficients and solve the Fokker–Planck equations are fully described in the "Methods" section of this paper. These calculations can be independently reproduced using any general-purpose programming language (e.g., Python, MATLAB, or Fortran) by implementing the equations provided.

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

## Acknowledgements

We greatly thank the NASA Juno mission team for providing the wave, particle, and magnetic field data. This study was supported by the National Natural Science Foundation of China under grant numbers 42025404 (B.N.), 42188101 (B.N.), and 423B2409 (M.L.), and by the National Key R&D Program of China (2022YFF0503700, B.N.), the Natural Science Foundation of Hubei Province, China (2025AFA030, B.N.), and the Xplorer prize (B.N.).

## Author contributions
B.N. and E.R. jointly oversaw the project and supervised the study at Wuhan University and the Max Planck Institute for Solar System Research, respectively. M.L. carried out the data analysis, conducted the numerical simulations, produced all figures, and wrote the initial draft of the manuscript under the co-supervision of B.N., E.R., and Q.M. B.N., E.R., Q.M., P.K., R.Z., G.C., N.K., X.C., P.L., Y.H., and S.W. contributed to the interpretation of results and refinement of the figures and manuscript. All authors participated in the discussion and interpretation of the data and reviewed the final manuscript. The authors affirm that all authors have contributed to the manuscript and uphold principles of inclusion, equity, and respect in global research collaborations.

## Funding

## Competing interests

The authors declare no competing interests.
