## [Transparent Peer Review file · Nature Communications]

A slot region in the magnetosphere of Jupiter

Corresponding Author: Dr Elias Roussos

Version 0:

Reviewer comments:

Reviewer #1

(Remarks to the Author)

This manuscript demonstrates a scenario to explain the depletion of electron fluxes in Jupiter's magnetosphere around the orbit of one of the Galilean satellites, Europa. The authors show that the electron fluxes in the energy range from tens of keV to a few hundreds of keV decrease around Europa's orbit, which is correlated well with the intense electromagnetic wave activity known as whistler-mode waves. Based on the data analysis and simulation results, the authors argue that the flux depletion around Europa's orbit results from the loss of electrons caused by wave-particle interaction between the electrons and the whistler-mode waves. In the Earth's magnetosphere, persistent whistler-mode wave activity forms the depletion of the relativistic electron fluxes, which is called the "slot region" of the radiation belt where relativistic electrons are contained around the Earth. The authors named the flux depletion region around Europa's orbit "a slot region" based on the analogy of the Earth's magnetosphere.

The manuscript is basically well-written with high readability. However, the reviewer thinks revisions are required before the manuscript is accepted for publication in Nature Communication. The reviewer hopes that the authors might consider to revise the manuscript following the comments described below.

Major comments

1. The flux depletion is observed in the energy range from tens to hundreds of keV. The energy range seems relatively low to argue "radiation belt" electrons in Jupiter's magnetosphere. The term "the slot region of Jupiter's radiation belt" catches the reader's eye, but the reviewer feels that the term does not reflect the contents in the manuscript. The reviewer hopes that the authors might consider rephrasing the term. Perhaps "A slot region of Jupiter's magnetosphere" may be more appropriate.
2. Line 54-55: The work done by Santolik et al. (2011) does not show the electron loss by whistler-mode waves. The electron absorption by Rhea forms electron loss-cone distribution, which contributes to the excitation of whistler-mode waves through temperature anisotropy arising from the loss-cone distribution.
3. The dawn-dusk asymmetry of the electron flux and whistler-mode wave power is quite interesting. Still, the reviewer is afraid of some biases in the statistical analysis performed by the authors. For example, as shown in the work by Shprits et al. (2018), wave intensity is much higher during the moon's flyby than the average wave power computed by excluding the periods of the moon's encounters. Suppose observations of Juno's dawn side passes were made during Europa's proximity compared with Juno's dusk side passes. In that case, the dawn-dusk asymmetry shown in the manuscript is possible. The reviewer hopes that the authors might consider to show the flux and wave power distributions, excluding the periods when the magnetic field line passing through Juno is in proximity to Europa. It is interesting to know how Juno is close to Europa when the observation in Figure 1 is made.
4. Line 239-250: Based on the mechanism proposed by the authors, electron fluxes should be sufficiently high enough to excite whistler-mode waves around Europa's orbit at the initial stage of the slot formation. However, the authors argue that electron fluxes and wave activity shown in Figure 1 are the typical features around Europa's orbit. The reviewer feels that the proposed mechanism is inconsistent with the observed features. Is the electron temperature anisotropy high enough to excite whistler-mode waves even under low electron fluxes? Are the Juno observations of electrons and waves assumed to be an equilibrium state around Europa's orbit?
5. Line 260-267: The reviewer feels that the "limited longitudinal extent of the electron slot" is related to the proximity to Europa rather than the magnetic local time. The reviewer hopes that the authors might clarify the fact that the slot formation depends on the magnetic local time rather than the proximity to Europa.

Minor comments

- Line 119: "The low p-value" may be "The p-value"

- Line 121: "at Jupiter" may be "at Europa's orbit"

Reviewer #2

(Remarks to the Author)

The authors presented the results of the Juno spacecraft's observation of whistler mode waves and energetic/relativistic electrons around the Europa orbit. The statistical analysis revealed strong whistler mode waves near Europa's orbit in the frequency range below $0.05 f_{ceq}$ on the dawn side of the magnetosphere. The authors showed the correlated relationship between whistler mode wave amplitudes and electron flux variations. The results of numerical simulations demonstrated that the observed wave amplitude is sufficient to explain the formation of the radiation belt slot near Europa's orbit within the time scale of hours. The manuscript is organized well. This reviewer can recommend publication after the authors have made some clarifications and revisions.

L104-108

The dawn-dusk asymmetry of the wave amplitude is identified. Does the distance from Europa not affect the wave amplitude distribution?

The authors related the clear depletion region for electron fluxes in the dawn sector to the dawn-dusk asymmetry of whistler mode wave amplitudes. Can the Fokker-Planck diffusion code reproduce the observed local time variation of electron fluxes?

L153-158

LFWs and HFWS can resonate electrons in the wide energy and pitch angle ranges. From the point of view of the wave generation mechanism, which energy and pitch angle ranges contribute to amplifying whistler mode waves to such a large amplitude?

Fig.4d

The electron flux around M-shell = 7 did not vary in the simulation because of the assumed boundary condition (fixed to the initial electron flux). Considering the radial diffusion process, how is the electron flux inward from Europa's orbit maintained?

L239-241

Which energy range of "injected hot electrons" is important for the excitation of whistler mode waves?

Reviewer #3

(Remarks to the Author)

Global comments:

The paper entitled "A slot region on Jupiter's radiation belt" is very interesting and quite well written. The subject of the article is really important in the modelling of Jupiter's radiation belts around Europa. The use of Juno data for this purpose (waves and particles) is new and of great interest. However, I suggest minor comments to improve the paper and the understanding for the reader.

Minor comments:

- Line 57: "M-shell" – I think you should specify here what magnetic field models are used. (I know that you mention it later in the paper).

- Line 64 and after: "Observations" – You do not mention Galileo waves data. Even if the paper is focused on Juno data, I think that one or two sentences are needed to mention them. Previous work using Galileo data showed the effect of waves on radiation belts.

- Lines 73-75: "Low frequency...intensities" - How did you differentiate between the two types of waves? Why $0.05f_{ceq}$?

- Line 75: "whistler-mode wave amplitudes" – On which frequency range the amplitude is integrated to obtain pT?

- Figure 3.h and 3.i: I do not understand if in those simulations only wave-particle interactions are taken into account. Is there also the radial diffusion? Is the decrease only linked to the scattering coefficients resulting from the wave-particle interaction?

- Line 171: "good agreement" – Good agreement is a bit vague. Could you please mention the ratio between observations and simulations?

- Figure 4d: Could you please add the points from observations?

- Figure 5: For which energy is the figure made?

- Line 267 : "synchrotron belts" – you mean radiation belts?

- Line 284: Why are the spectra corrected?

- Line 296: "converted" – What do you mean?? Do you use local magnetic field measurements and then calculate equatorial magnetic field using the magnetic field lines defined by JRM33 + CON2020? Wouldn't it be better to use the location of the spacecraft and then calculate local AND equatorial magnetic field with JRM33+CON2020? This avoids mixing measured and calculated data for the magnetic field.

- Lines 315 and after: Do you use the wave frequency spectrums directly in the code that calculate the diffusion coefficients or do you fit them with Gaussian functions? For the computations of diffusion coefficients, how many harmonics do you considered? For the wave normal distribution, I do not understand where the peak values and the angular width come from. It seems that the LFW are not field aligned at all at high MLAT, why?

- Line 345 and after : Usually in Fokker-Planck diffusion simulation, the cross terms from wave-particle interaction are not taken into account for numerical reasons. Is it the case here?

- Line 351: Isn't there a mistake in the formula? I don't think there's a "p" in the denominator at the very beginning of the formula.

- Line 360 : What does the constant 3.325×10^{-8} correspond to?

- Line 371: what does "hybrid" means?

- Lines 401-404: why do you choose to use the electron pitch angle diffusion rates at M-shell=9.5 and not at the centered of the range where waves are located (M-shell=8-10)? Why do you mean by "the electron PSD remains constant at both the lower and upper M-shell boundaries? Constant in time? Why do you use D&G model for the initial electron flux and not Juno data?

- Line 443: The author's name is missing a 'K'.

Version 1:

Reviewer comments:

Reviewer #1

(Remarks to the Author)

The authors kindly consider the comments raised by the reviewer, and revised the manuscript. The revision made by the authors address the reviewer's concerns, which makes the reviewer satisfied. The reviewer thinks that the revised manuscript can be recommended for the publication in Nature Communications.

(The title displayed in the system does not match the one in the revised manuscript, but the reviewer believes that this discrepancy will be corrected during the publication process.)

Reviewer #2

(Remarks to the Author)

This reviewer examined the authors' replies and found that their revisions were reasonable. Additional analyses clarified the significance of the results and strengthened the conclusion of the manuscript. This reviewer can recommend the publication of the revised manuscript.

Reviewer #3

(Remarks to the Author)

I would like to thank the authors for their clear responses to the reviewers' comments and the consequent changes to the manuscript. So, in my opinion, the manuscript is publishable as it stands.

We sincerely thank the reviewers and the editor for thorough and important reviews of our manuscript, as well as for the constructive comments and valuable suggestions. We greatly appreciate all the time and efforts that the reviewers have dedicated to evaluating our work. In response to all the comments provided, we have conducted more work, carefully revised the manuscript and addressed each point in detail to improve the clarity and quality of the manuscript. In this reply letter, we provide point-by-point responses to the comments and suggestions from all the reviewers.

Responses to Reviewer #1

This manuscript demonstrates a scenario to explain the depletion of electron fluxes in Jupiter's magnetosphere around the orbit of one of the Galilean satellites, Europa. The authors show that the electron fluxes in the energy range from tens of keV to a few hundreds of keV decrease around Europa's orbit, which is correlated well with the intense electromagnetic wave activity known as whistler-mode waves. Based on the data analysis and simulation results, the authors argue that the flux depletion around Europa's orbit results from the loss of electrons caused by wave-particle interaction between the electrons and the whistler-mode waves. In the Earth's magnetosphere, persistent whistler-mode wave activity forms the depletion of the relativistic electron fluxes, which is called the "slot region" of the radiation belt where relativistic electrons are contained around the Earth. The authors named the flux depletion region around Europa's orbit "a slot region" based on the analogy of the Earth's magnetosphere.

The manuscript is basically well-written with high readability. However, the reviewer thinks revisions are required before the manuscript is accepted for publication in Nature Communication. The reviewer hopes that the authors might consider to revise the manuscript following the comments described below.

Reply:

We greatly appreciate the reviewer for the careful review of our manuscript and for the constructive comments and suggestions below. We are grateful for the basically positive evaluation from the reviewer and have addressed his/her comments in the revised version of the manuscript. Our point-by-point responses are provided below.

1. The flux depletion is observed in the energy range from tens to hundreds of keV. The energy range seems relatively low to argue "radiation belt" electrons in Jupiter's magnetosphere. The term "the slot region of Jupiter's radiation belt" catches the reader's eye, but the reviewer feels that the term does not reflect the contents in the manuscript. The reviewer hopes that the authors might consider rephrasing the term. Perhaps "A slot region of Jupiter's magnetosphere" may be more appropriate.

Reply:

We thank the reviewer for this constructive suggestion to improve the clarity of our manuscript. We fully agree that the energy range of our focus in this manuscript—from tens to hundreds of keV—is lower than the typical relativistic electron energies ($>\sim$ MeV) commonly associated with Jupiter's radiation belts. At Earth, the slot region of electron fluxes is not only observed for several MeV energies, but also for the energy range from \sim 30 keV to 1 MeV (Reeves et al., 2003; Ma et al., 2016; Claudepierre et al., 2020). The electrons at tens to hundreds of keV are the "seed" population

for local acceleration to multiple MeV energies in the terrestrial magnetosphere (Thorne et al., 2013; Jaynes et al., 2015) and Jupiter (Shprits et al., 2012; Ni et al., 2018; Ma et al., 2024). Our study specifically targets on the “seed” electron population in Jupiter’s inner magnetosphere, revealing a slot region for the electron fluxes at energies of tens to hundreds of keV. In light of the reviewer’s constructive comment, we have revised the manuscript accordingly and rephrased the title to “A slot region of Jupiter’s magnetosphere”, as the reviewer has suggested, to more accurately reflect the scope and content of our work.

In addition, we also add some necessary material in the revised manuscript to clarify this point. Please see Lines 76-78 in the change-tracked version. “*The electrons at tens to hundreds of keV are the “seed” population for local acceleration to multiple MeV energies in the magnetospheres of Earth (Thorne et al., 2013; Jaynes et al., 2015) and Jupiter (Ma et al., 2024).*”

References:

1. Reeves, G. D., McAdams, K. L., Friedel, R. H. W. & O’Brien, T. P. Acceleration and loss of relativistic electrons during geomagnetic storms. *Geophys. Res. Lett.* **30**, 1529 (2003).
2. Shprits, Y. Y., Menietti, J. D., Gu, X., Kim, K. C. & Horne, R. B. Gyroresonant interactions between the radiation belt electrons and whistler mode chorus waves in the radiation environments of Earth, Jupiter, and Saturn: A comparative study. *J. Geophys. Res. Space Phys.* **117**, A11216 (2012).
3. Ni, B., Huang, J. et al. Importance of electron distribution profiles to chorus wave driven evolution of Jovian radiation belt electrons. *Earth Planet. Phys.* **2**, 371–383 (2018).
4. Ma, Q. et al. Characteristic energy range of electron scattering due to plasmaspheric hiss. *J. Geophys. Res. Space Phys.* **121**, 11,737–11,749 (2016).
5. Claudepierre, S. G. et al. Empirically estimated electron lifetimes in the Earth’s radiation belts: Comparison with theory. *Geophys. Res. Lett.* **47**, e2019GL086056 (2020).
6. Thorne, R. M. et al. Rapid local acceleration of relativistic radiation-belt electrons by magnetospheric chorus. *Nature.* **504**, 411–414 (2013).
7. Jaynes, A. N. et al. Source and seed populations for relativistic electrons: Their roles in radiation belt changes. *J. Geophys. Res. Space Phys.* **120**, 7240–7254 (2015).
8. Ma, Q. et al. Generation and impacts of whistler-mode waves during energetic electron injections in Jupiter’s outer radiation belt. *J. Geophys. Res. Space Phys.* **129**, e2024JA032624 (2024).

2. Line 54-55: *The work done by Santolík et al. (2011) does not show the electron loss by whistler-mode waves. The electron absorption by Rhea forms electron loss-cone distribution, which contributes to the excitation of whistler-mode waves through temperature anisotropy arising from the loss-cone distribution.*

Reply:

We thank the reviewer for pointing out this important aspect and for the insightful comment. After revisiting the work of Santolík et al. (2011) and the related studies, we provide a more nuanced clarification as below.

The study by Santolík et al. (2011) emphasizes that the absorption of electrons at the surface of Rhea leads to the formation of electron loss-cone distributions, which in turn can generate temperature anisotropies that excite intense whistler-mode waves. We fully agree with the reviewer that the work of Santolík et al. (2011) does not show electron loss caused by whistler-mode waves.

Meanwhile, we would like to note that these excited whistler-mode waves—which can propagate obliquely—are considered to be responsible for spatially extended losses of energetic electrons on the flanks of Rhea, beyond the immediate absorption region. This broader electron depletion, initially attributed to possible dust interactions, can be explained by the presence of whistler-mode waves, as discussed in the later literature (e.g., Roussos et al., 2012).

Moreover, these spatially extended losses were observed to propagate downstream of Rhea, although not too far (typically within $\sim 1^\circ$ in longitude). Therefore, while Santolík et al. (2011) primarily demonstrated the generation of whistler waves via the temperature anisotropy associated with loss-cone distributions, this work can support the idea that these waves can subsequently induce local and near-downstream energetic electron losses.

In response to the reviewer's valuable suggestion, we have carefully revised the relevant sentence in the manuscript. The original sentence: “*Whistler-mode wave driven electron loss has also been observed at the Saturnian moons, such as Rhea*” has now been modified to be: “*At Saturn’s moons, such as Rhea, strong whistler-mode waves are excited by the electron loss-cone distributions, which may further contribute to spatially extended losses of energetic electrons observed in the vicinity of the moons* (Santolík et al., 2011; Roussos et al., 2012).” Please see Lines 55-58 in the revised manuscript with tracked changes.

We think that this revised description can reflect more accurately the physical processes identified in the previous literature and address the reviewer’s comment. Again, we are grateful for this comment from the reviewer, which indeed helps us improve the clarity and accuracy of our manuscript.

References:

1. Santolík, O. et al. Intense plasma wave emissions associated with Saturn’s moon Rhea. *Geophys. Res. Lett.* **38**, L19204 (2011).
2. Roussos, E. et al. Energetic electron observations of Rhea’s magnetospheric interaction. *Icarus*. **221**, 116-134 (2012)

3. *The dawn-dusk asymmetry of the electron flux and whistler-mode wave power is quite interesting. Still, the reviewer is afraid of some biases in the statistical analysis performed by the authors. For example, as shown in the work by Shprits et al. (2018), wave intensity is much higher during the moon’s flyby than the average wave power computed by excluding the periods of the moon’s encounters. Suppose observations of Juno’s dawn side passes were made during Europa’s proximity compared with Juno’s dusk side passes. In that case, the dawn-dusk asymmetry shown in the manuscript is possible. The reviewer hopes that the authors might consider to show the flux and wave power distributions, excluding the periods when the magnetic field line passing through Juno is in proximity to Europa. It is interesting to know how Juno is close to Europa when the observation in Figure 1 is made.*

Reply:

We thank the reviewer for this insightful and constructive comment regarding the analysis method for our statistical study, which is crucial to improve the quality of our study. To carefully address this important comment, we have performed further investigation following a five-step procedure described as follows:

(1) Clarification on Europa Flyby Data Exclusion

In this study, we have utilized Juno observations from Perijove 03 (PJ-03) to Perijove 46 (PJ-

46) to investigate the global distributions of energetic electrons and whistler-mode waves in Jupiter’s magnetosphere. Following the reviewer’s comment, we have re-examined the Juno orbital data and confirmed that there was only one close flyby of Europa during the entire period of PJ03 – PJ46, which occurred during PJ45 (Kurth et al., 2023). After re-checking our dataset, we realize that the Europa flyby data has already been excluded from our analysis, although this was not clearly stated in the original manuscript. We sincerely apologize for this omission. In the revised manuscript, we have now explicitly clarified this point (Lines 114-115): “*Note the Europa flyby data during PJ-45 are excluded from our analysis to avoid any potential effect of moon proximity.*” We appreciate the reviewer for highlighting this important point.

(2) Evaluation of Europa’s Proximity Effect

To further investigate the reviewer’s comment regarding the moon proximity effect, we have selected all wave data within M-shells of 7 – 12 and examined the dependence of wave amplitude on (a) the distance between Juno and Europa (normalized by Europa’s radius, R_E) and (b) the M-shell value at the magnetic latitudes between 10° – 60° . We note that due to the data limitation on the dawnside at the magnetic latitudes below 10° , we have focused our analysis on the latitudes between 10° – 60° with much more samples.

As shown in **Figure R1**, (a) the wave amplitudes on the dawnside (00-06 MLT) are generally higher than those on the duskside (18-24 MLT), consistent with our statistical findings. (b) Interestingly, duskside wave samples tend to occur closer to Europa than the dawnside samples. Therefore, the observed dawn-dusk asymmetry in whistler-mode wave amplitudes cannot be a result of closer proximity of Juno’s dawnside passes to Europa.

In panel (b) of Figure R1, a peak in wave amplitude is observed near M-shell ~ 9.5 on the dawnside, while a duskside peak appears near M-shell ~ 8.5 . When the M-shell of Juno deviates from the M-shell of Europa, the wave amplitudes tend to decrease gradually. This analysis reinforces that the observed asymmetries are intrinsic rather than a by-product of Europa’s proximity.

Figure R1. Whistler-mode ave amplitudes as functions of (a) the distance between Juno and Europa, and (b) the M-shell, on both the dawnside and duskside, for observations at magnetic latitudes between 10° – 60° . The data are averaged over every $5R_E$ in distance and every 1 in M-shell, respectively.

(3) Magnetic Local Time Separation Analysis

The longitude or magnetic local time (MLT) separation can provide more important

information than the absolute radial distance when evaluating the spatial relationship between Europa and wave amplitude observations. Radial distance can be significantly influenced by latitudinal differences between Juno and Europa, even when they are located at similar M-shells and longitudes. Variations in magnetic latitude can lead to large differences in radial distance, making it a potentially misleading parameter for interpreting physical processes.

Hence, we present the whistler-mode wave amplitude samples as a function of the magnetic local time difference between Juno and Europa (i.e., $|\text{MLT}_{\text{Juno}} - \text{MLT}_{\text{Europa}}|$) on the dawnside and duskside across four different M-shell regions (**Figure R2**). The wave amplitudes do not show a monotonic trend with increasing local time interval, indicating that the observed dawn-dusk asymmetry in whistler-mode wave amplitude is not simply due to closer proximity of Juno's passes to Europa on the dawnside.

Although no clear dependence of wave amplitude on the MLT separation from Europa is observed over the full orbit, it remains possible that Europa's influence is stronger at small angular separations but becomes averaged out at larger distances along the orbit. This behavior is similar to what has been observed for Io (Clark et al., 2023; Kotsiaros et al., 2024), where the strongest perturbations are detected within several tens of degrees downstream, while further downstream the dependence on Io's location becomes weak or negligible. Therefore, it is important to consider that the localized Europa-related effect may still be present, even if it is not easily distinguishable in a global statistical sense.

These findings provide further evidence that the reported dawn-dusk asymmetry in whistler-mode wave amplitude is not an observational artifact related to Europa.

Figure R2. Wave amplitudes as a function of magnetic local time (MLT) separation between Juno and Europa for four different M-shell regions, at magnetic latitudes between 10° and 60° , on

both the dawn and dusk sides. The MLT data are averaged over 2-hour intervals.

(4) Consideration of Different Wave Types

We acknowledge the work by Shprits et al. (2018), which showed that whistler-mode wave intensities significantly increase near Europa and Ganymede, particularly associated with chorus waves. To further investigate the global distribution of Jovian chorus waves at frequencies between $0.1f_{ceq}$ and f_{ceq} (f_{ceq} is the equatorial electron gyrofrequency), Lu et al. (2024) combined Juno data during PJ01–PJ56 and Galileo observations and found that the most intense chorus waves are generally located at M-shells between 8 and 11, near the magnetic equator and predominantly on the duskside. However, in our analysis, we have considered all the whistler-mode wave activity below the equatorial electron cyclotron frequency, including both chorus-like and hiss-like components (Ma et al., 2024). The hiss-like waves contribute significantly to the wave powers at the magnetic latitudes of 20° – 50° , particularly on the dawnside. As a result, our statistical analysis reveals relatively higher average wave amplitudes on the dawnside than duskside, contrasting with previous studies that focused primarily on chorus waves.

These additional analyses support the robustness of our original conclusions and confirm that the observed dawn-dusk asymmetry in whistler-mode wave amplitude is a real phenomenon other than an artifact caused by Europa’s proximity.

(5) Clarification of Juno-Europa Orbital Configuration during Figure 1 Observation

To further address the reviewer’s comment, we have revised the description of Juno’s orbital configuration during the observation event of Figure 1, which is shown in **Figure R3** (now **Figure S1** in the supplementary materials). During this orbit, intense whistler-mode waves were observed from 20:00 to 22:30 UT. At 21:37 UT, Juno crossed the same magnetic field line as that Europa had passed earlier at 20:00 UT. This temporal and spatial proximity indicates that the observed wave enhancements are likely related to localized coupling processes between Europa and Jupiter’s magnetosphere, rather than a broad, global effect. This clarification further supports that while Europa can locally modify the surrounding plasma environment and wave activity, the global asymmetry patterns observed in Jupiter’s magnetosphere are largely independent of Europa’s immediate influence. We have added some necessary explanation. Please see Lines 10-19 in the revised Supplementary. In addition, we have made the revision in the revised manuscript. Please see Lines 90-91. “Notably, at 21:37 UT the Juno spacecraft passed the same magnetic field line (M-shell~9.6) as that Europa crossed earlier at 20:00 UT (Supplementary Fig. S1).”

Figure R3. Juno and Europa trajectories in the Jupiter magnetic (JM) coordinate on July 10, 2017.

Again, we thank the reviewer for this constructive and important comments, which greatly helps us to significantly improve the quality and clarity of our manuscript.

References:

1. Kurth, W. S. et al. Juno plasma wave observations at Europa. *Geophys. Res. Lett.* **50**, e2023GL105775 (2023).
2. Clark, G. et al. Energetic proton acceleration by EMIC waves in Io's footprint tail. *Front. Astron. Space. Sci.* **10**, 2023 (2023).
3. Kotsiaros, S. et al. Juno observations set new constraints on the electrodynamic interaction between Io and Jupiter. *J. Geophys. Res. Space Phys.* **129**, e2024JA032591 (2024).
4. Shprits, Y. Y. et al. Strong whistler mode waves observed in the vicinity of Jupiter's moons. *Nat. Commun.* **9**, 3131 (2018).
5. Lu, P., Cao, X., Ni, B., Wang, S. & Long, M. Statistical distribution of chorus waves in Jupiter's magnetosphere based on Galileo and Juno observations. *J. Geophys. Res. Planets* **130**, e2024JE008818 (2025).
6. Ma, Q. et al. Survey of whistler-mode wave amplitudes and frequency spectra in Jupiter's magnetosphere. *Geophys. Res. Lett.* **51**, e2024GL111882 (2024).

4. Line 239-250: Based on the mechanism proposed by the authors, electron fluxes should be sufficiently high enough to excite whistler-mode waves around Europa's orbit at the initial stage of the slot formation. However, the authors argue that electron fluxes and wave activity shown in Figure 1 are the typical features around Europa's orbit. The reviewer feels that the proposed mechanism is inconsistent with the observed features. Is the electron temperature anisotropy high enough to excite whistler-mode waves even under low electron fluxes? Are the Juno observations of electrons and waves assumed to be an equilibrium state around Europa's orbit?

Reply:

We sincerely appreciate the reviewer for providing this important comment and raising questions regarding the consistency between our proposed mechanism and the observed features of electrons and waves. As a matter of fact, the excitation of whistler-mode waves depends on a number of factors, including the ambient magnetic field strength, background electron density, hot electron flux, and temperature anisotropy. Previous studies (e.g., Ma et al., 2024) have shown that the temperature anisotropy of injected electrons with energies from a few keV to ~50 keV can effectively drive the growth of whistler-mode waves in Jupiter's magnetosphere. While the wave excitation is generally more efficient under high fluxes of source electron population, enhanced temperature anisotropy can still generate whistler-mode waves even when the overall electron flux is relatively low. Positive wave growth rate requires sufficient temperature anisotropy near the resonance energy of electrons. Around Europa's orbit, such high anisotropy is expected to be acquirable due to Europa's interaction with the magnetospheric plasma. For example, Europa absorbs electrons at lower equatorial pitch angles (Long et al., 2022), reducing the parallel electron fluxes and thereby increasing the temperature anisotropy. This enhanced anisotropy can provide more free energy for the wave growth even under conditions of low electron fluxes. In this way, it is reasonable that Juno can often observe the activity of whistler-mode waves accompanied by low electron fluxes of a few to tens of keV around Europa's orbit.

We interpret the wave activity and electron flux distributions shown in Figure 1 as a quasi-steady state, reflecting the cumulative effect of repeated injections and sustained wave-particle

interactions over time. While the observed features represent a quasi-steady state of Jupiter's inner magnetosphere, they do not necessarily indicate an equilibrium condition since the local injections and wave-particle interactions continuously modulate the system. As illustrated in Figure 1b, the energy-time spectrum of electron fluxes shows dispersive signatures resulting from electron injections. Meanwhile, intense whistler-mode waves were observed, and the electron fluxes from tens to hundreds of keV decreased at the magnetic latitudes of $\sim 22^\circ$ – 42° during this interval. Correspondingly, Juno did not observe the electrons with high equatorial pitch angles. Instead, the electron fluxes at low equatorial pitch angles probably decrease to support the enhancement of temperature anisotropy near the equator.

While our current simulation can explain the general formation of the energetic electron slot under the impact of whistler-mode waves near Europa, we fully agree with the reviewer that further analysis and modeling attempts are needed to fully understand the evolution of this slot region with more details, which will be a subject of our follow-up investigation. We have added this point to the main text. Please see Lines 285-290 in the revised manuscript with tracked changes.

References:

1. Ma, Q. et al. Generation and impacts of whistler-mode waves during energetic electron injections in Jupiter's outer radiation belt. *J. Geophys. Res. Space Phys.* **129**, e2024JA032624 (2024).
2. Long, M., Ni, B., Cao, X., Gu, X. & Kollmann, P. Losses of radiation belt energetic particles by encounters with four of the inner moons of Jupiter. *J. Geophys. Res. Planets* **127**, e2021JE007050 (2022).

5. Line 260-267: *The reviewer feels that the "limited longitudinal extent of the electron slot" is related to the proximity to Europa rather than the magnetic local time. The reviewer hopes that the authors might clarify the fact that the slot formation depends on the magnetic local time rather than the proximity to Europa.*

Reply:

We thank the reviewer for this valuable comment, which is closely related to the above Comment #3 raised by the reviewer regarding the dawn-dusk asymmetry of the electron flux and whistler-mode wave power in the observations. According to our further analysis in this regard and our detailed response to Comment #3, we think that the limited spatial extent of the electron slot is primarily related to the magnetic local time rather than the longitudinal proximity to Europa. To clarify this point, we have revised the corresponding text as follows to avoid any potential confusion. Please see Lines 285–288 in the revised manuscript: *"In addition, the formation of the electron slot is characterized by a limited extent in magnetic local time, with the most prominent depletion observed in the dawn sector. This feature indicates that the slot is predominantly strongly influenced by the MLT-dependent wave activity"*.

Line 119: *"The low p-value" may be "The p-value"*

Line 121: *"at Jupiter" may be "at Europa's orbit"*

Reply:

We thank the reviewer for these suggested changes. We have revised the text accordingly to improve the descriptions. Please see Lines 134 and 136.

Responses to Reviewer #2

The authors presented the results of the Juno spacecraft's observation of whistler mode waves and energetic/relativistic electrons around the Europa orbit. The statistical analysis revealed strong whistler mode waves near Europa's orbit in the frequency range below $0.05 f_{ceq}$ on the dawn side of the magnetosphere. The authors showed the correlated relationship between whistler mode wave amplitudes and electron flux variations. The results of numerical simulations demonstrated that the observed wave amplitude is sufficient to explain the formation of the radiation belt slot near Europa's orbit within the time scale of hours. The manuscript is organized well. This reviewer can recommend publication after the authors have made some clarifications and revisions.

Reply:

We greatly appreciate the reviewer for the careful review of our manuscript and for the constructive comments and suggestions below. We are grateful for the positive evaluation from the reviewer and have addressed his/her comments in the revised version of the manuscript. Our point-by-point responses are provided below.

L104-108

The dawn-dusk asymmetry of the wave amplitude is identified. Does the distance from Europa not affect the wave amplitude distribution?

Reply:

We sincerely thank the reviewer for this insightful and constructive comment regarding the analysis method for our statistical study, which is crucial to improve the quality of our study. Actually, this comment is very similar to Comment #3 from Reviewer #1 (as shown above). To carefully address this important comment, we have performed the following further investigation:

(1) Clarification on Europa Flyby Data Exclusion

In this study, we have utilized Juno observations from Perijove 03 (PJ-03) to Perijove 46 (PJ-46) to investigate the global distributions of energetic electrons and whistler-mode waves in Jupiter's magnetosphere. Following the reviewer's comment, we have re-examined the Juno orbital data and confirmed that there was only one close flyby of Europa during the entire period of PJ03 – PJ46, which occurred during PJ45 (Kurth et al., 2023). After re-checking our dataset, we realize that the Europa flyby data has already been excluded from our analysis, although this was not clearly stated in the original manuscript. We sincerely apologize for this omission. In the revised manuscript, we have now explicitly clarified this point (Lines 114-115): “*Note the Europa flyby data during PJ-45 are excluded from our analysis to avoid any potential effect of moon proximity.*”

(2) Evaluation of Europa's Proximity Effect

To further investigate the reviewer's comment regarding the moon proximity effect, we have selected all wave data within M-shells of 7 – 12 and examined the dependence of wave amplitude on (a) the distance between Juno and Europa (normalized by Europa's radius, R_E) and (b) the M-shell value at the magnetic latitudes between 10° – 60° . We note that due to the data limitation on the dawnside at the magnetic latitudes below 10° , we have focused our analysis on the latitudes between 10° – 60° with much more samples.

As shown in **Figure R1**, (a) the wave amplitudes on the dawnside (00-06 MLT) are generally higher than those on the duskside (18-24 MLT), consistent with our statistical findings. (b) Interestingly, duskside wave samples tend to occur closer to Europa than the dawnside samples.

Therefore, the observed dawn-dusk asymmetry in whistler-mode wave amplitudes cannot be a result of closer proximity of Juno's dawnside passes to Europa.

In panel (b) of Figure R1, a peak in wave amplitude is observed near M-shell ~ 9.5 on the dawnside, while a duskside peak appears near M-shell ~ 8.5 . When the M-shell of Juno deviates from the M-shell of Europa, the wave amplitudes tend to decrease gradually. This analysis reinforces that the observed asymmetries are intrinsic rather than a by-product of Europa's proximity.

Figure R1. Whistler-mode ave amplitudes as functions of (a) the distance between Juno and Europa, and (b) the M-shell, on both the dawnside and duskside, for observations at magnetic latitudes between 10° - 60° . The data are averaged over every $5R_E$ in distance and every 1 in M-shell, respectively.

(3) Magnetic Local Time Separation Analysis

The longitude or magnetic local time (MLT) separation can provide more important information than the absolute radial distance when evaluating the spatial relationship between Europa and wave amplitude observations. Radial distance can be significantly influenced by latitudinal differences between Juno and Europa, even when they are located at similar M-shells and longitudes. Variations in magnetic latitude can lead to large differences in radial distance, making it a potentially misleading parameter for interpreting physical processes.

Hence, we present the whistler-mode wave amplitude samples as a function of the magnetic local time difference between Juno and Europa (i.e., $|\text{MLT}_{\text{Juno}} - \text{MLT}_{\text{Europa}}|$) on the dawnside and duskside across four different M-shell regions (**Figure R2**). The wave amplitudes do not show a monotonic trend with increasing local time interval, indicating that the observed dawn-dusk asymmetry in whistler-mode wave amplitude is not simply due to closer proximity of Juno's passes to Europa on the dawnside.

Although no clear dependence of wave amplitude on the MLT separation from Europa is observed over the full orbit, it remains possible that Europa's influence is stronger at small angular separations but becomes averaged out at larger distances along the orbit. This behavior is similar to what has been observed for Io (Clark et al., 2023; Kotsiaros et al., 2024), where the strongest perturbations are detected within several tens of degrees downstream, while further downstream the dependence on Io's location becomes weak or negligible. Therefore, it is important to consider that the localized Europa-related effect may still be present, even if it is not easily distinguishable in a

global statistical sense.

These findings provide further evidence that the reported dawn-dusk asymmetry in whistler-mode wave amplitude is not an observational artifact related to Europa.

Figure R2. Wave amplitudes as a function of magnetic local time (MLT) separation between Juno and Europa for four different M-shell regions, at magnetic latitudes between 10° and 60° , on both the dawn and dusk sides. The MLT data are averaged over 2-hour intervals.

(4) Consideration of Different Wave Types

We acknowledge the work by Shprits et al. (2018), which showed that whistler-mode wave intensities significantly increase near Europa and Ganymede, particularly associated with chorus waves. To further investigate the global distribution of Jovian chorus waves at frequencies between $0.1 f_{ceq}$ and f_{ceq} (f_{ceq} is the equatorial electron gyrofrequency), Lu et al. (2024) combined Juno data during PJ01–PJ56 and Galileo observations and found that the most intense chorus waves are generally located at M-shells between 8 and 11, near the magnetic equator and predominantly on the duskside. However, in our analysis, we have considered all the whistler-mode wave activity below the equatorial electron cyclotron frequency, including both chorus-like and hiss-like components (Ma et al., 2024). The hiss-like waves contribute significantly to the wave powers at the magnetic latitudes of 20° – 50° , particularly on the dawnside. As a result, our statistical analysis reveals relatively higher average wave amplitudes on the dawnside than duskside, contrasting with previous studies that focused primarily on chorus waves.

These additional analyses support the robustness of our original conclusions and confirm that the observed dawn-dusk asymmetry in whistler-mode wave amplitude is a real phenomenon other than an artifact caused by Europa’s proximity.

(5) Clarification of Juno-Europa Orbital Configuration during Figure 1 Observation

Furthermore, we have carefully checked Juno’s orbital configuration during the observation

event of Figure 1, which is shown in **Figure R3** (now **Figure S1** in the supplementary materials). During this orbit, intense whistler-mode waves were observed from 20:00 to 22:30 UT. At 21:37 UT, Juno crossed the same magnetic field line as that Europa had passed earlier at 20:00 UT. This temporal and spatial proximity indicates that the observed wave enhancements are likely related to localized coupling processes between Europa and Jupiter’s magnetosphere, rather than a broad, global effect. This clarification further supports that while Europa can locally modify the surrounding plasma environment and wave activity, the global asymmetry patterns observed in Jupiter’s magnetosphere are largely independent of Europa’s immediate influence. We have added some necessary explanation. Please see Lines 10-19 in the revised Supplementary. In addition, we have made the revision in the revised manuscript. Please see Lines 90-91. “Notably, at 21:37 UT the Juno spacecraft passed the same magnetic field line (M -shell~9.6) as that Europa crossed earlier at 20:00 UT (Supplementary Fig. S1).”

Figure R3. Juno and Europa trajectories in the Jupiter magnetic (JM) coordinate on July 10, 2017.

Again, we thank the reviewer for this constructive and important comments, which greatly helps us to significantly improve the quality and clarity of our manuscript.

References:

1. Kurth, W. S. et al. Juno plasma wave observations at Europa. *Geophys. Res. Lett.* **50**, e2023GL105775 (2023).
2. Clark, G. et al. Energetic proton acceleration by EMIC waves in Io’s footprint tail. *Front. Astron. Space. Sci.* **10**, 2023 (2023).
3. Kotsiaros, S. et al. Juno observations set new constraints on the electrodynamic interaction between Io and Jupiter *J. Geophys. Res. Space Phys.* **129**, e2024JA032591 (2024).
4. Shprits, Y. Y. et al. Strong whistler mode waves observed in the vicinity of Jupiter’s moons. *Nat. Commun.* **9**, 3131 (2018).
5. Lu, P., Cao, X., Ni, B., Wang, S. & Long, M. Statistical distribution of chorus waves in Jupiter’s magnetosphere based on Galileo and Juno observations. *J. Geophys. Res. Planets* **130**, e2024JE008818 (2025).
6. Ma, Q. et al. Survey of whistler-mode wave amplitudes and frequency spectra in Jupiter’s magnetosphere. *Geophys. Res. Lett.* **51**, e2024GL111882 (2024).

The authors related the clear depletion region for electron fluxes in the dawn sector to the dawn-dusk asymmetry of whistler mode wave amplitudes. Can the Fokker-Planck diffusion code reproduce the observed local time variation of electron fluxes?

Reply:

We greatly appreciate the reviewer for this valuable comment. Compared to the dawn sector, the average amplitudes of whistler-mode waves on the duskside are generally weaker. Although the electron flux depletion (i.e., the electron slot) in the dusk sector is not as pronounced as in the dawn sector, the overall electron flux levels in the dusk sector are higher than the dawnside. This may suggest additional sources in the dusk sector to replenish electrons in the slot region around Europa.

In order to investigate the local time effect for the sake of reproducing the observed local time variation of electron fluxes, a four-dimensional (4-D) Fokker - Planck diffusion simulation, which includes the (M-shell, MLT, energy and pitch angle) dependencies of wave-induced electron diffusion and scattering rates, is needed to simulate the dawn-dusk asymmetry of electron fluxes in Jupiter's magnetosphere. To do so, detailed information of the global profiles of the wave power spectrum, convective electric field, background magnetic field strength and electron density, etc, is required, which however is insufficiently available at this stage.

Therefore, with available data, so far we have not implemented the 4-D but 2-D and 1-D Fokker-Planck diffusion simulations to model the spatial variations of electron fluxes in this study. We provide some reasonable qualitative explanations for the dawn-dusk asymmetry in electron flux, i.e., a combined effect of large-scale dawn-dusk electric fields and asymmetric whistler wave amplitude distributions. Our analysis based on the pure radial diffusion simulation (Fig. 4d) indicates that, with the currently estimated diffusion coefficients, the whistler wave-induced losses can lead to the formation of a depletion region. The less distinct slot observed on the duskside implies the presence of additional acceleration mechanisms or a stronger-than-expected inward radial diffusion in the dusk magnetosphere. We have also discussed this issue in the Supplementary. Please see Lines 22-32 in the Supplementary. *“The wave amplitudes are stronger on the dawnside compared to the duskside, consistent with the more pronounced depletion of electron fluxes in the dawn sector. The observed dawn-dusk asymmetries in both electron fluxes and whistler-mode waves are likely influenced by the large-scale dawn-dusk electric field³⁸⁻⁴². This electric field drives inward transport of electrons to lower M-shells on the dayside and outward transport on the nightside. Electrons accelerated by inward transport on the dayside drift in Jupiter's rotational direction, leading to increased electron fluxes in the dusk sector and a corresponding decrease in the dawn sector. Meanwhile, the combination of the large-scale dawn-dusk electric field with the globally asymmetric distribution of whistler-mode wave amplitudes can further facilitate the observed profile of the dawn-dusk asymmetry in electron fluxes, which however requires four-dimensional (4-D) Fokker - Planck diffusion simulations for further exploration.”*

L153-158

LFWs and HFWs can resonate electrons in the wide energy and pitch angle ranges. From the point of view of the wave generation mechanism, which energy and pitch angle ranges contribute to amplifying whistler mode waves to such a large amplitude?

Reply:

We thank the reviewer for this valuable question. It concerns the specific energy and pitch angle ranges of electrons that can contribute to the amplification of whistler-mode waves to large

amplitude. The growth of whistler-mode waves, including both the low-frequency (LFWs) and high-frequency components (HFWs), is primarily driven by temperature anisotropy ($T_{\perp} > T_{\parallel}$) of the energetic electron population. According to recent studies (e.g., Ma et al., 2024), injected electrons with energies ranging from a few keV to ~ 50 keV can provide sufficient free energy through anisotropic distributions to excite whistler-mode chorus waves (HFWs) in Jupiter’s magnetosphere. The resonant electron energies for LFWs are somehow higher. Previous Juno observations suggested that the energies of injecting electrons could reach above 300 keV (e.g., Haggerty et al., 2017). Near the equator, the pitch angle distributions of injecting electrons around Europa are usually pancake-like (Tomás et al., 2004; Ma et al., 2021). If the anisotropic electron fluxes are high, the waves can also be amplified at lower frequencies. From the perspective of pitch angle, electrons with equatorial pitch angles between $\sim 60^{\circ}$ and 90° tend to contribute significantly to the wave growth. These electrons are more likely to exhibit strong perpendicular temperature components (T_{\perp}), which are essential to drive the cyclotron resonance instability of whistler-mode waves.

The high temperature anisotropy can also be induced by electron collisional losses near the magnetic conjunction with Europa (Shprits et al., 2018). As Jupiter’s dipole tilt rotates, the moon is a few degrees away from the magnetic equator. The electrons with low and middle pitch angles could be absorbed by Europa while the $\sim 90^{\circ}$ equatorial pitch angle electrons are not affected, inducing high anisotropy favorable for the cyclotron resonance instability. The moon absorption effect is more evident for higher energy electrons than lower energy electrons, because the bounce period of higher energy electrons is shorter while the drift period is close to the corotation period (Long et al., 2022). Hence, LFWs are probably amplified by the high anisotropy of high-energy electrons after their collisional loss by the moon.

Overall, HFWs and LFWs can resonate with ~ 1 -50 keV electrons and $> \sim 50$ keV electrons, respectively, over the nearly entire pitch angle range. The source electron population at equatorial pitch angles of $\sim 60^{\circ}$ - 90° with anisotropic distributions is significant to the wave generation and amplification. During the injection events, the source electron fluxes can be large for the wave growth. Even when the source electron fluxes are lower, increased electron temperature anisotropy induced by electron collisional losses can also provide free energy to drive the cyclotron resonance instability of waves. To respond, we have clarified this point in the main text. Please see Lines 260-266 in the change-tracked version: “*HFWs and LFWs can resonate with ~ 1 -50 keV electrons and $> \sim 50$ keV electrons, respectively, over the nearly entire pitch angle range. The source electron population at equatorial pitch angles of $\sim 60^{\circ}$ - 90° with anisotropic distributions is significant to the wave generation and amplification. It has been shown that injected electrons with typical energies of a few keV to ~ 50 keV and occasionally reaching above 300 keV, exhibiting temperature anisotropies, are capable of exciting whistler-mode chorus waves under typical Jovian magnetospheric conditions (Ma et al., 2024).*”

References:

1. Ma, Q. et al. Generation and impacts of whistler-mode waves during energetic electron injections in Jupiter’s outer radiation belt. *J. Geophys. Res. Space Phys.* **129**, e2024JA032624 (2024).
2. Haggerty, D. K. et al. Jovian injections observed at high latitude. *Geophys. Res. Lett.* **46**, 9397–9404 (2019).
3. Tomás, A. T. et al. Energetic electrons in the inner part of the Jovian magnetosphere and their relation to auroral emissions. *J. Geophys. Res. Space Phys.* **109**, A06203 (2004).

4. Ma, Q. et al. Energetic electron distributions near the magnetic equator in the Jovian plasma sheet and outer radiation belt using Juno observations. *Geophys. Res. Lett.* **48**, e2021GL095833 (2021).
5. Shprits, Y. Y. et al. Strong whistler mode waves observed in the vicinity of Jupiter's moons. *Nat. Commun.* **9**, 3131 (2018).
6. Long, M., Ni, B., Cao, X., Gu, X. & Kollmann, P. Losses of radiation belt energetic particles by encounters with four of the inner moons of Jupiter. *J. Geophys. Res. Planets* **127**, e2021JE007050 (2022).

Fig.4d The electron flux around M-shell = 7 did not vary in the simulation because of the assumed boundary condition (fixed to the initial electron flux). Considering the radial diffusion process, how is the electron flux inward from Europa's orbit maintained?

Reply:

We thank the reviewer for this important question regarding the boundary condition and its influence on the simulated electron flux near M-shell = 7. As pointed out by the reviewer, in our simulation the electron flux at M~7 remains unchanged due to the imposed inner boundary condition, which is set as the initially electron flux provided by one model and fixed throughout the simulation. We agree that under realistic conditions, radial diffusion could transport electrons inward from the outer magnetosphere or from Europa's orbit to lower M-shells (Woodfield et al., 2014), potentially changing the electron population near M~7. We would like to note that since the primary focus of our simulation is to investigate the effect of local wave-particle interactions in the spatial extent of M-shell = 8 - 11, we have adopted the above fixed boundary values as a common approach to constrain the system and simplify interpretation of the wave-driven effects included in the radial diffusion simulation with loss terms (see equation (9) in the Methods section).

Such a treatment can be roughly justified by comparing the respective timescales of the three considered physical processes: radial diffusion (RD), Europa's absorption (EA), and wave-driven loss (WL). The timescale of radial diffusion at M-shell = 7 is about 25 Earth days, inferred from the radial diffusion coefficient (see equation (10) in the Methods section). This timescale is significantly longer than that of electron pitch angle scattering loss due to whistler-mode waves (~hours) but comparable to the collisional loss timescale induced by Europa (~tens of days). As a result, the radial diffusion only is unlikely to vary largely the electron phase space densities at M-shell = 7 within the timescale of electron loss near Europa (several hours). Furthermore, it is expected that variations at the simulation boundary of M-shell = 7 on the timescales of tens of days has very small impacts on our simulation results for which the wave-driven loss is dominant with the timescales of a few hours. Meanwhile, we acknowledge that incorporating dynamic boundary conditions or global radial diffusion from the outer regions may improve the reliability of our simulations, which will be considered in future work.

To respond to this important comment, we have added some materials to explain why we take such a constant boundary condition. Please see Lines 440-444 in the change-tracked version: “By considering that the radial diffusion timescale of tens of days (equation (10)) is significantly longer than that of electron pitch angle scattering loss due to whistler-mode waves (~hours) but comparable to the collisional loss timescale induced by Europa (~tens of days), we further set that the electron PSD remains constant at both the lower (M-shell = 7) and upper M-shell boundaries (M-shell = 12) so that the effect of wave-driven loss can be reasonably assessed.”

Reference:

1. Woodfield, E. E., Horne, R. B., Glauert, S. A., Menietti, J. D. & Shprits, Y. Y. The origin of Jupiter's outer radiation belt. *J. Geophys. Res. Space Phys.* **119**, 3490–3502 (2014).

L239-241 Which energy range of "injected hot electrons" is important for the excitation of whistler mode waves?

Reply:

We thank the reviewer for this valuable question. As a matter of fact, the excitation of whistler-mode waves is influenced by a number of factors, including the ambient magnetic field strength, background electron density, hot electron fluxes, and the temperature anisotropy of the source electron population. Recent studies (e.g., Ma et al., 2024) have demonstrated that injected electrons with typical energies of a few keV to ~50 keV and occasionally reaching above 300 keV, when exhibiting sufficient temperature anisotropy (T_{\perp}/T_{\parallel}), can effectively drive the growth of whistler-mode chorus waves in Jupiter's magnetosphere. To respond, we have clarified this point in the revised manuscript to improve its completeness. Please see Lines 263-266 in the change-tracked version. "*It has been shown that injected electrons with typical energies of a few keV to ~50 keV and occasionally reaching above 300 keV, exhibiting temperature anisotropies, are capable of exciting whistler-mode chorus waves under typical Jovian magnetospheric conditions (Ma et al., 2024).*"

Reference:

1. Ma, Q. et al. Generation and impacts of whistler-mode waves during energetic electron injections in Jupiter's outer radiation belt. *J. Geophys. Res. Space Phys.* **129**, e2024JA032624 (2024).

Responses to Reviewer #3

Global comments:

The paper entitled “A slot region on Jupiter’s radiation belt” is very interesting and quite well written. The subject of the article is really important in the modelling of Jupiter’s radiation belts around Europa. The use of Juno data for this purpose (waves and particles) is new and of great interest. However, I suggest minor comments to improve the paper and the understanding for the reader.

Reply:

We greatly appreciate the reviewer for the careful review of our manuscript and for the constructive comments and suggestions below. We are grateful for the positive evaluation from the reviewer and have addressed his/her comments in the revised version of the manuscript. Our point-by-point responses are provided below.

Minor comments:

- Line 57: “M-shell” – I think you should specify here what magnetic field models are used. (I know that you mention it later in the paper).

Reply:

We thank the reviewer for this valuable comment. We have revised the text accordingly. Please see Lines 61-63 in the change-tracked version: “*In this study, the M-shell parameter is calculated based on the JRM33 internal magnetic field model (order 13) and the CON2020 current sheet model to characterize the magnetic geometry in our analysis.*”

- Line 64 and after: “Observations” – You do not mention Galileo waves data. Even if the paper is focused on Juno data, I think that one or two sentences are needed to mention them. Previous work using Galileo data showed the effect of waves on radiation belts.

Reply:

We thank the reviewer for this valuable comment and suggestion. We have now added a brief discussion of the Galileo wave observations to acknowledge their important contributions to our understanding of wave–particle interactions in Jupiter’s radiation belts. Although our analysis focuses on Juno wave data, we agree that the earlier findings based on Galileo observations are highly relevant and provide important context for this study.

The following sentence has been added at the beginning of the Observations section. Please see Lines 71-73: “*Previous studies using Galileo wave data have demonstrated the significant role of plasma waves in shaping Jupiter’s radiation belts, especially through wave–particle interactions that can lead to both particle acceleration and loss (e.g., Menietti et al., 2012, 2016).*”

References:

1. Menietti, J. D. *et al.* Chorus, ECH, and Z mode emissions observed at Jupiter and Saturn and possible electron acceleration. *J. Geophys. Res. Space Phys.* **117**, A12214 (2012).
2. Menietti, J. D. *et al.* Survey of whistler mode chorus intensity at Jupiter. *J. Geophys. Res. Space Phys.* **121**, 9758–9770 (2016).

- Lines 73-75: “Low frequency...intensities” - How did you differentiate between the two types of waves? Why 0.05fceeq?

Reply:

We thank the reviewer for this valuable comment regarding the classification of wave types and the choice of $0.05f_{ceq}$ as the separation between them. Following the approach used in Ma et al. (2024), we differentiate between chorus-like and hiss-like whistler-mode waves based on a spectral gap consistently observed in Juno data near $0.05f_{ceq}$, where f_{ceq} denotes the electron gyrofrequency at the magnetic equator. In their statistical study, the authors found that chorus waves are predominantly observed in the frequency range of $0.05-1f_{ceq}$ closely following the variation of f_{ceq} , and often exhibit discrete elements. In contrast, hiss-like waves are mainly observed at frequencies below $0.05f_{ceq}$, with a relatively weaker dependence on the local gyrofrequency and a more diffuse spectral structure.

This spectral separation becomes particularly evident at M-shells less than 10, where the chorus and hiss populations are morphologically distinct. The use of $0.05f_{ceq}$ as the boundary is therefore supported by the observational evidence and allows for a consistent classification scheme.

Furthermore, this classification is physically meaningful. Higher-frequency chorus waves ($f > 0.05f_{ceq}$) are more likely to be locally generated, while lower-frequency hiss-like waves ($f < 0.05f_{ceq}$) at $M < 10$ may result from the wave propagation from remote sources, as suggested by Wang et al. (2008). This approach is also consistent with previous studies in Earth's magnetosphere, where chorus waves are typically found above $0.05f_{ceq}$ (Li et al., 2016).

In order to clarify this, we have add necessary materials in our revised manuscript. Please see Lines 81-85 in the change-tracked version: “Following the previous work (Ma et al., 2024), we categorize the whistler-mode waves at $0.05f_{ceq} < f < f_{ceq}$ frequencies as chorus or high frequency whistler-mode waves (HFW) and the waves at $f < 0.05f_{ceq}$ as hiss waves or low frequency whistler-mode waves (LFW) (Fig.1a). LFWs with frequencies below $0.05f_{ceq}$ exhibited strong wave power, while HFWs above $0.05f_{ceq}$ displayed relatively weaker intensities.”

References:

1. Ma, Q. et al. Survey of whistler-mode wave amplitudes and frequency spectra in Jupiter' s magnetosphere. *Geophys. Res. Lett.* **51**, e2024GL111882 (2024).
2. Wang, K., Thorne, R. M. & Horne, R. B. Origin of Jovian hiss in the extended Io torus. *Geophys. Res. Lett.* **35**, L16105 (2008)..
3. Li, W. et al. New chorus wave properties near the equator from Van Allen Probes wave observations. *Geophys. Res. Lett.* **43**, 4725–4735 (2016).

-Line 75: “whistler-mode wave amplitudes” – On which frequency range the amplitude is integrated to obtain pT?

Reply:

We thank the reviewer for this valuable comment. The wave amplitudes in picotesla (pT) are integrated over the entire frequency range identified as whistler-mode waves (i.e., 50 Hz to f_{ceq}). The specific criteria for identifying the whistler-mode waves are described in detail in the Methods section. Please see Lines 332-334: “(2) The wave spectra observed with frequencies between 50 Hz and the equatorial electron gyrofrequency (f_{ceq}) are analyzed to cover a broad frequency range for whistler-mode wave identification.”

To respond to this question, we have added a sentence in the main text to clarify the calculation of whistler-mode wave amplitudes. Please see Lines 85-87 in the change-tracked version: “We calculate the whistler-mode wave amplitudes over the frequency range from 50 Hz to the equatorial

electron gyrofrequency (f_{ceq}) (see Methods).”

-Figure 3.h and 3.i: I do not understand if in those simulations only wave-particle interactions are taken into account. Is there also the radial diffusion? Is the decrease only linked to the scattering coefficients resulting from the wave-particle interaction?

Reply:

We appreciate the reviewer for this valuable comment. In Figures 3h and 3i, the simulations are performed at fixed L-shells (or M-shells) to specifically examine the effects of wave-particle interactions only. That is, the radial diffusion is not included in these simulations. As a result, the modeled decrease of electron phase space density is solely due to pitch angle scattering by whistler-mode waves. In contrast, we examine the effect of radial diffusion in our following simulation runs, as illustrated in Figure 4d. As we see from these figures, the wave-driven scattering can result in much more rapid electron losses compared to the radial diffusion, therefore dominating the occurrence of the electron slot in Jupiter’s inner magnetosphere.

-Line 171: “good agreement” – Good agreement is a bit vague. Could you please mention the ratio between observations and simulations?

Reply:

We thank the reviewer for this valuable comment. Following the reviewer’s suggestion to provide a more quantitative assessment, we have evaluated the ratios of the averaged electron phase space densities between the simulations and observations for the considered three energy channels. The ratios are approximately 0.95, 0.95, and 0.98, respectively, indicating that the model captures the overall trend of the measured electron fluxes with good agreement.

To respond to this comment, we have added the above point in the main text. Please see Lines 186-189 in the change-tracked version: “*The simulation results show reasonable consistency with the observations, with the ratios of the averaged electron phase space densities between the simulations and observations for the three indicated energy channels being approximately 0.95, 0.95, and 0.98, indicating that the model captures the overall trend of the measured electron fluxes.*”

-Figure 4d: Could you please add the points from observations?

Reply:

We greatly appreciate the reviewer’s suggestion and fully understand the value of directly comparing simulation results with observational data in Figure 4d. We have made efforts in this direction, but several practical limitations prevent us from reliably adding observational points to the figure at this stage. As described below, the main issue is the difference between the Juno observations and the Divine & Garrett model of electron fluxes, the latter of which is used as the initial and boundary conditions in our simulations to produce Figure 4.

Firstly, to run the radial diffusion simulations, we require reasonable initial conditions for electron distributions as a function of both energy and equatorial pitch angle. However, Juno’s electron observations are insufficient to provide the full set of data needed. For example, the JEDI-E instrument measures electron fluxes in the energy range from ~30 keV to ~1 MeV, typically across 24 energy channels. Since our simulation solves the radial diffusion equation at fixed first and second adiabatic invariants, we require a broader and smoother energy-pitch angle distribution than that Juno can currently provide. As illustrated in Fig R4, we show the variations of Jupiter’s

magnetospheric electron energy as a function of L-shell for a number of fixed values of the first adiabatic invariant. It is clear that the electron energy changes largely with L-shell. Therefore, adoption of JEDI-E statistical electron distribution model should involve extensive interpolation across both energy and pitch angle, which will undoubtedly introduce much uncertainty into the simulations. In contrast, although the Divine & Garrett (D&G) model only provides a parameterized distribution of > 60 keV electron fluxes as a function of energy and equatorial pitch angle, the resulting electron flux distributions are smoother and more continuous, making them more suitable for numerical simulations. By considering that the main purpose of our simulations is to illustrate that the moon absorption alone cannot account for the formation of the energetic electron slot but inclusion of wave-driven electron scattering can, usage of the D&G model distribution is sufficient for our quantitative investigation.

Figure R4. Variations of electron energy as a function of L-shell for the considered fixed values of the first adiabatic invariant.

However, we also would like to note that Juno observed omnidirectional electron fluxes (e.g., at ~ 97 keV) are around 1–2 orders of magnitude higher than the D&G model results in the same region of Jupiter’s magnetosphere. This discrepancy highlights the need for further data analyses and the development of updated energetic electron models for Jupiter’s magnetosphere, which is beyond the scope of the current study. We agree that a direct comparison of simulations with observations should be valuable. We plan to incorporate more Juno-based observational constraints in future work as more data become available and more robust electron flux models are under development. One promising direction is to combine data from both JADE-E (~ 0.1 -30 keV) and JEDI-E (~ 30 keV–1 MeV) instruments to construct a more comprehensive model of energetic electron distribution in Jupiter’s magnetosphere. However, careful cross-calibration between these two datasets will be essential to ensure consistency and reliability.

-Figure 5: For which energy is the figure made?

Reply:

We thank the reviewer for this comment. In our study, we primarily focus on electrons in the energy range of tens to hundreds of keV. Figure 5 is presented as a conceptual illustration to help visualize the physical processes that contribute to the formation of the energetic electron slot near Europa. It does not correspond to a specific energy channel but instead summarizes the overall physical processes derived from our investigation.

-Line 267: “synchrotron belts” – you mean radiation belts?

Reply:

Thanks. We have changed “synchrotron belts” into “radiation belts” to ensure accurate terminology.

-Line 284: Why are the spectra corrected?

Reply:

We thank the reviewer for this comment. The spectra are corrected due to the contamination caused by minimum ionizing peaks, which typically appear at energies around 100-200 keV. These artifacts result from penetrating high-energy electrons that produce spurious counts in the detector. We have revised the text to clarify this point. Please see Lines 313-316 in the change-tracked version: “The electron energy spectra were corrected according to the method proposed by Mauk et al.⁴³ by removing artifacts caused by minimum ionizing peaks (~100-200 keV) that result from the contamination by penetrating particles.”

-Line 296: “converted” – What do you mean?? Do you use local magnetic field measurements and then calculate equatorial magnetic field using the magnetic field lines defined by JRM33 + CON2020? Wouldn't it be better to use the location of the spacecraft and then calculate local AND equatorial magnetic field with JRM33+CON2020? This avoids mixing measured and calculated data for the magnetic field.

Reply:

We thank the reviewer for this valuable comment. We agree that the term “converted” may be misleading. We have replaced it with “mapped” for clarity.

In our approach, we use the measured local magnetic field for mapping to the equatorial magnetic field strength along the same field line with the JRM33+CON2020 magnetic model. Specifically, the magnetic fields from JRM33 + CON2020 model at the Juno location and at the equator are $B_{1,local}$ and $B_{1,eq}$, and the magnetic field measured by Juno is $B_{0,local}$. The equatorial magnetic field is calculated as $B_{0,eq} = B_{0,local} * B_{1,eq} / B_{1,local}$. This method is more consistent with the actual observational conditions, as it incorporates real magnetic field measurements rather than relying solely on the model field values at the spacecraft location.

Moreover, this approach is also necessary for calculating the wave-induced bounce-averaged electron diffusion coefficients, which requires the knowledge of the magnetic field distribution along the field line. Therefore, a combination of using the JRM33+CON2020 model and mapping the observed magnetic field to the equator is appropriate for our research purpose in this study. We have made the necessary revision to clarify this point. Please see Lines 326-329 in the change-tracked version: “The local magnetic field measurements provided by the Juno magnetometer (MAG)(Connerney et al., 2017) are mapped to the magnetic equator to estimate the equatorial field strength (B_{eq}) along the same field line using the JRM33+CON2020 model(Connerney et al., 2020,

2022).”

References:

1. Connerney, J. E. P. et al. The Juno magnetic field investigation. *Space Sci. Rev.* 213, 39–138 (2017).
2. Connerney, J. E. P. et al. A new model of Jupiter’s magnetic field at the completion of Juno’s prime mission. *J. Geophys. Res. Planets.* 127, e2021JE007055 (2022).
3. Connerney, J. E. P., Timmins, S., Hecceg, M. & Joergensen, J. L. A Jovian magnetodisc model for the Juno era. *J. Geophys. Res. Space Phys.* 125, e2020JA028138 (2020)

-Lines 315 and after: Do you use the wave frequency spectrums directly in the code that calculate the diffusion coefficients or do you fit them with Gaussian functions? For the computations of diffusion coefficients, how many harmonics do you considered? For the wave normal distribution, I do not understand where the peak values and the angular width come from. It seems that the LFW are not field aligned at all at high MLAT, why?

Reply:

We sincerely thank the reviewer for this important comment and a number of detailed questions regarding the wave spectral input, harmonic resonance, and wave normal angle distribution used in our diffusion coefficient calculations.

Firstly, we have adopted the observations of wave frequency spectra directly in the computation of electron diffusion coefficients, without fitting them into any Gaussian function. The wave power spectra are taken from our statistical results, thereby preserving the actual spectral characteristics observed by Juno. This information has been provided in our original manuscript. Please see Lines 375-376 in the revised manuscript.

Secondly, we have considered the harmonic resonances from $N = -10$ to $+10$, including both the cyclotron and Landau resonance, to ensure that all potentially relevant wave-particle interactions for electrons in the targeted energy range are covered. This information has been provided in our original manuscript. Please see Lines 377-379 in the revised manuscript.

Thirdly, regarding the wave normal angle distribution, we have adopted a parameterized model following previous studies in Earth’s magnetosphere (Bortnik et al., 2011; Ni et al., 2013), in which the waves are close to field-aligned near the equator and gradually become oblique as they propagate to higher latitudes. At each latitude, the wave normal angles are assumed to follow a Gaussian distribution centered at a specified peak value with a given angular width. In our study, the peak values and angular widths are selected following the previous studies mentioned above. This approach is necessary due to the current lack of comprehensive wave normal angle measurements from Juno. This information has been provided in our original manuscript. Please see Lines 365-367, Supplementary Table 1, and cited references #47-49 in the revised manuscript.

Specifically, concerning the behavior of low-frequency whistler-mode waves (LFWs) at high magnetic latitudes, we agree with the reviewer that these waves appear less field-aligned in this region. This is supported by recent ray tracing and observational studies (Kang et al., 2025; Ma et al., 2024), which show that LFWs, particularly at high latitudes, tend to develop larger propagation angles with respect to the background magnetic field. This behavior arises due to the field line geometry, wave refraction effects and the influence of plasma density gradients along the magnetic field lines.

As illustrated in Figure R5 (adapted from Kang et al. (2025)), the ray tracing results reveal that

high-frequency waves, which are typically generated near the magnetic equator, are largely confined within $\pm 20^\circ$ in magnetic latitude due to the Landau damping. In contrast, low-frequency waves can propagate to higher latitudes and lower M-shells. During this process, their wave normal angles increase progressively with magnetic latitude, which can clearly explain why LFWs at high latitudes are often oblique.

Figure R5. (a) All the ray paths with color representing wave normal angle and transparency representing wave power (identical to Figure 1(b)), but in the M-shell and magnetic latitude coordinates, (b) the same format as Panel (a), but only showing the high frequency whistler-mode (HFW) waves. The black dots on the upper right corner are the endpoints of some rays, and (c) the same format as Panel (a), but only showing the low frequency whistler-mode (LFW) waves. The black dots on the upper right corner are the ends of some rays, which are identical with those in Panel (b), showing that these rays undergo transition from LFW to HFW at these locations. (adopted from Kang et al. (2025))

To respond to this important comment and answer the questions, we have added the relevant material in the main text for clarification. Please see Lines 359-363 in the change-tracked version: “Ray tracing results show that high-frequency whistler-mode waves generated near the magnetic equator of Jupiter’s magnetosphere are typically confined within $|MLAT| < \sim 20^\circ$, while low-frequency waves can reach higher latitudes. As these waves propagate, their wave normal angles become increasingly oblique due to the effects of field line geometry, plasma density gradient and wave refraction.”

References:

1. Bortnik, J., Chen, L., Li, W., Thorne, R. M., Meredith, N. P., & Horne, R. B. Modeling the wave power distribution and characteristics of plasmaspheric hiss. *J. Geophys. Res.* **116**, A12209 (2011).
2. Ni, B., Bortnik, J., Thorne, R. M., Ma, Q. & Chen, L. Resonant scattering and resultant pitch angle evolution of relativistic electrons by plasmaspheric hiss. *J. Geophys. Res. Space Phys.* **118**, 7740–7751 (2013).
3. Ma, Q. et al. Survey of whistler-mode wave amplitudes and frequency spectra in Jupiter’s magnetosphere. *Geophys. Res. Lett.* **51**, e2024GL111882 (2024).
4. Kang, N., Ma, Q., Bortnik, J., Qin, M. & Li, W. Ray tracing of whistler mode waves in Jupiter’s magnetosphere. *Geophys. Res. Lett.* **52**, e2024GL113727 (2025).

-Line 345 and after: Usually in Fokker-Planck diffusion simulation, the cross terms from wave-particle interaction are not taken into account for numerical reasons. Is it the case here?

Reply:

We thank the reviewer for this valuable comment. Yes, the cross diffusion terms are often omitted in Fokker–Planck diffusion simulations to ensure numerical stability, particularly when their contributions are relatively minor.

In our study, however, we have included the cross terms. Although the cross terms are not important when pitch angle scattering is the dominate process, they can becomes important when the momentum diffusion or electron acceleration plays a role (e.g., Subbotin et al., 2010). We have clarified this point in the revised manuscript. Please see Lines 391-392 in the revised manuscript.

Reference:

1. Subbotin, D., Shprits, Y. & Ni, B. Three-dimensional VERB radiation belt simulations including mixed diffusion. *J. Geophys. Res. Space Phys.* **115**, A03205 (2010).

-Line 351: Isn't there a mistake in the formula? I don't think there's a "p" in the denominator at the very beginning of the formula.

Reply:

We appreciate the reviewer's careful reading. The formula indeed has multiple equivalent forms depending on how the phase space density (PSD) is expressed. We have carefully checked the equation and confirm that it is correct as written in our manuscript. The form we use is consistent with Equation (1) in Xiao et al. (2009). We have added the citation in the text to clarify this point for readers.

Reference:

1. Xiao, F., Su, Z., Zheng, H. & Wang, S. Modeling of outer radiation belt electrons by multidimensional diffusion process. *J. Geophys. Res. Space Phys.* **114**, A03201 (2009).

-Line 360: What does the constant 3.325×10^{-8} correspond to?

Reply:

We thank the reviewer for this comment. After careful re-calculation, we find that the coefficient should be 3.33×10^{-8} . The constant 3.33×10^{-8} is a unit conversion factor used in the transformation between electron flux and phase space density (PSD). Please check below for the details of the equation deduction.

The electron momentum (p) is related to the kinetic energy (E) as

$$p^2 c^2 = E^2 + 2E m_0 c^2$$

where c is the speed of light and m_0 is the electron restmass, The electron phase space density f is calculated from flux j as

$$f = \frac{j}{p^2} = \frac{j c^2}{E^2 + 2E m_0 c^2}$$

Now we consider the unit conversion.

$$\begin{aligned} f &= \frac{j[\text{cm}^{-2}\text{s}^{-1}\text{sr}^{-1}\text{keV}^{-1}] \cdot c^2}{(E[\text{MeV}])^2 + 2E[\text{MeV}] \cdot m_0 c^2[\text{MeV}]} \cdot \frac{\text{cm}^{-2}\text{s}^{-1}\text{keV}^{-1}}{(\text{MeV})^2} \\ &= \frac{j[\text{cm}^{-2}\text{s}^{-1}\text{sr}^{-1}\text{keV}^{-1}]}{(E[\text{MeV}])^2 + 2E[\text{MeV}] \cdot m_0 c^2[\text{MeV}]} \cdot \left(\frac{c}{\text{MeV} \cdot \text{cm}}\right)^3 \cdot \frac{\text{cm} \cdot \text{s}^{-1}}{c} \cdot \frac{\text{MeV}}{\text{keV}} \\ &= \frac{j[\text{cm}^{-2}\text{s}^{-1}\text{sr}^{-1}\text{keV}^{-1}]}{(E[\text{MeV}])^2 + 2E[\text{MeV}] \cdot m_0 c^2[\text{MeV}]} \cdot \left(\frac{c}{\text{MeV} \cdot \text{cm}}\right)^3 \cdot \frac{\text{cm} \cdot \text{s}^{-1}}{3 \times 10^{10} \text{cm} \cdot \text{s}^{-1}} \times 10^3 \\ &= 3.33 \times 10^{-8} \frac{j[\text{cm}^{-2}\text{s}^{-1}\text{sr}^{-1}\text{keV}^{-1}]}{(E[\text{MeV}])^2 + 2E[\text{MeV}] \cdot m_0 c^2[\text{MeV}]} \cdot \left(\frac{c}{\text{MeV} \cdot \text{cm}}\right)^3 \end{aligned}$$

Accordingly, we have replaced “3.325” by “3.33” in the revised manuscript. Please see equation (7). The corresponding formulation can be available from previous references (Ni et al., 2009; Cervantes et al., 2020).

References:

1. Ni, B. et al. Reanalyses of the radiation belt electron phase space density using nearly equatorial CRRES and polar-orbiting Akebono satellite observations. *J. Geophys. Res. Space. Phys.* **114**, A05208 (2009).
2. Cervantes, S. et al. Identifying radiation belt electron source and loss processes by assimilating spacecraft data in a three-dimensional diffusion model. *J. Geophys. Res. Space. Phys.* **125**, e2019JA027514 (2020).

-Line 371: what does “hybrid” means?

Reply:

We thank the reviewer for this comment. The term “hybrid” here refers to a specific numerical scheme used to solve the Fokker–Planck partial differential equation. This approach combines different finite-difference schemes to ensure both numerical stability and accuracy when handling the diffusion terms. We follow the method described in Xiao et al. (2009), where the hybrid scheme is introduced to optimize the solution of the Fokker–Planck equation under complex wave–particle interaction conditions. The Fokker Planck equation (equation (6)) includes the pitch angle diffusion term ($\langle D_{\alpha\alpha} \rangle$), momentum diffusion term ($\langle D_{pp} \rangle$), and the mixed terms ($\langle D_{\alpha p} \rangle$) and ($\langle D_{p\alpha} \rangle$). During each time step of calculation, we use fully implicit numerical method to solve the pitch angle diffusion and momentum terms, and alternating implicit numerical method to solve the mixed terms. Such a combination of solving methods is called “hybrid”.

Reference:

1. Xiao, F., Su, Z., Zheng, H. & Wang, S. Modeling of outer radiation belt electrons by multidimensional diffusion process. *J. Geophys. Res. Space Phys.* **114**, A03201 (2009).

-Lines 401-404: why do you choose to use the electron pitch angle diffusion rates at M-shell=9.5 and not at the centered of the range where waves are located (M-shell=8-10)? Why do you mean by “the electron PSD remains constant at both the lower and upper M-shell boundaries? Constant in time? Why do you use D&G model for the initial electron flux and not Juno data?

Reply:

We greatly appreciate the reviewer for this important comment.

Firstly, the observation event in Figure 1 is shown at M-shell~9.5. Correspondingly, we have selected M-shell = 9.5 as a representative location within the range of M-shell = 8–10 for calculating electron pitch angle diffusion rates because the wave intensity does not show significant variation across this range according to our statistical analysis. This serves as a reasonable approximation for illustrating the effects of wave–particle interactions within the slot region.

Secondly, the statement that “the electron PSD remains constant at both the lower and upper M-shell boundaries” refers to a time-invariant boundary condition applied during the simulation. This assumption allows us to isolate and evaluate the impact of local pitch angle scattering by whistler-mode waves within the modeled region. In reality, Jupiter’s radiation belts are dynamic, and we lack sufficient observational constraints outside the simulation domain to apply more realistic time-dependent boundary conditions. Using fixed boundary values is a common approach

in studies of radiation belts or remote planetary systems such as Jupiter and Saturn, where continuous, well-resolved particle measurements are limited. Such a treatment can also be roughly justified by comparing the respective timescales of the three considered physical processes: radial diffusion (RD), Europa's absorption (EA), and wave-driven loss (WL). By considering that the radial diffusion timescale of tens of days (equation (10)) is significantly longer than that of electron pitch angle scattering loss due to whistler-mode waves (~hours) but comparable to the collisional loss timescale induced by Europa (~tens of days), we further set that the electron PSD remains constant at both the lower (M-shell = 7) and upper M-shell boundaries (M-shell = 12) so that the effect of wave-driven loss can be reasonably assessed.

Lastly, the initial electron flux profile is derived from the Divine and Garrett (D&G) model because it provides a smooth and complete reference distribution across the relevant energy and spatial ranges. Juno's data, while highly valuable, are spatially and temporally limited and may not offer a complete or unbiased initial condition for the simulation. Using a model-based input helps to reduce uncertainties in this regard.

-Line 443: The author's name is missing a 'K'.

Reply:

Thanks. We have corrected it.